# Cross-compartment signal propagation in the mitotic exit network

**Xiaoxue Zhou[1]\*, Wenxue Li[2], Yansheng Liu[2], Angelika Amon[1]\***

[1]David H. Koch Institute for Integrative Cancer Research, Howard Hughes Medical Institute, Massachusetts Institute of Technology, Cambridge, United States; [2]Yale Cancer Biology Institute, Department of Pharmacology, Yale University, West Haven, United States

**Abstract** In budding yeast, the mitotic exit network (MEN), a GTPase signaling cascade, integrates spatial and temporal cues to promote exit from mitosis. This signal integration requires transmission of a signal generated on the cytoplasmic face of spindle pole bodies (SPBs; yeast equivalent of centrosomes) to the nucleolus, where the MEN effector protein Cdc14 resides. Here, we show that the MEN activating signal at SPBs is relayed to Cdc14 in the nucleolus through the dynamic localization of its terminal kinase complex Dbf2-Mob1. Cdc15, the protein kinase that activates Dbf2-Mob1 at SPBs, also regulates its nuclear access. Once in the nucleus, priming phosphorylation of Cfi1/Net1, the nucleolar anchor of Cdc14, by the Polo-like kinase Cdc5 targets Dbf2-Mob1 to the nucleolus. Nucleolar Dbf2-Mob1 then phosphorylates Cfi1/Net1 and Cdc14, activating Cdc14. The kinase-primed transmission of the MEN signal from the cytoplasm to the nucleolus exemplifies how signaling cascades can bridge distant inputs and responses.

**\*For correspondence:**
xiaoxue@mit.edu (XZ);
angelika@mit.edu (AA)

**Competing interests:** The authors declare that no competing interests exist.

## Introduction

In cellular signaling, the sensing of signals (e.g. binding of signaling molecules at cell surface) and the response (e.g. transcription in the nucleus) often occur in different cellular compartments. Determining how signals are transmitted across compartments is thus essential for understanding signal transmission. The mitotic exit network (MEN), a Ras-like GTPase kinase signaling cascade and budding yeast homolog of the Hippo pathway (*Hergovich and Hemmings, 2012*), represents such an example for signaling across cellular compartments. The MEN-activating signal is sensed and processed at the cytoplasmic face of spindle pole bodies (SPBs; yeast functional equivalent of the centrosomes), whereas the MEN effector protein Cdc14 resides in the nucleolus (*Figure 1A*). Because budding yeast undergoes a closed mitosis without disassembling the nuclear envelope and nucleolus, the MEN must transmit a signal generated at the cytoplasmic face of SPBs, across the nuclear envelope and into the nucleolus to activate its effector Cdc14. The molecular mechanisms governing this cross-compartment signaling process remain largely unknown.

The central function of the MEN is to couple the final cell cycle transition, exit from mitosis (when the mitotic spindle is disassembled, chromosomes decondense and cytokinesis ensues), to nuclear/spindle position. In many organisms including budding yeast, fission yeast, and some plant species, the site of cytokinesis/division plane (i.e. the bud neck) is determined prior to mitosis (*Guertin et al., 2002*). Thus, the mitotic spindle in these organisms must be positioned appropriately to ensure accurate genome partitioning between the daughter cells. In addition, these organisms have evolved surveillance mechanisms to monitor spindle position and delay cell cycle progression in response to mispositioned spindles. This surveillance mechanism is best understood in budding yeast where spindle position controls the activity of the MEN.

The MEN senses spindle position through a Ras-like GTPase Tem1. Tem1 is activated when a SPB enters the bud (*Bardin et al., 2000*; *Pereira et al., 2000*). Together with the Polo-like kinase Cdc5,

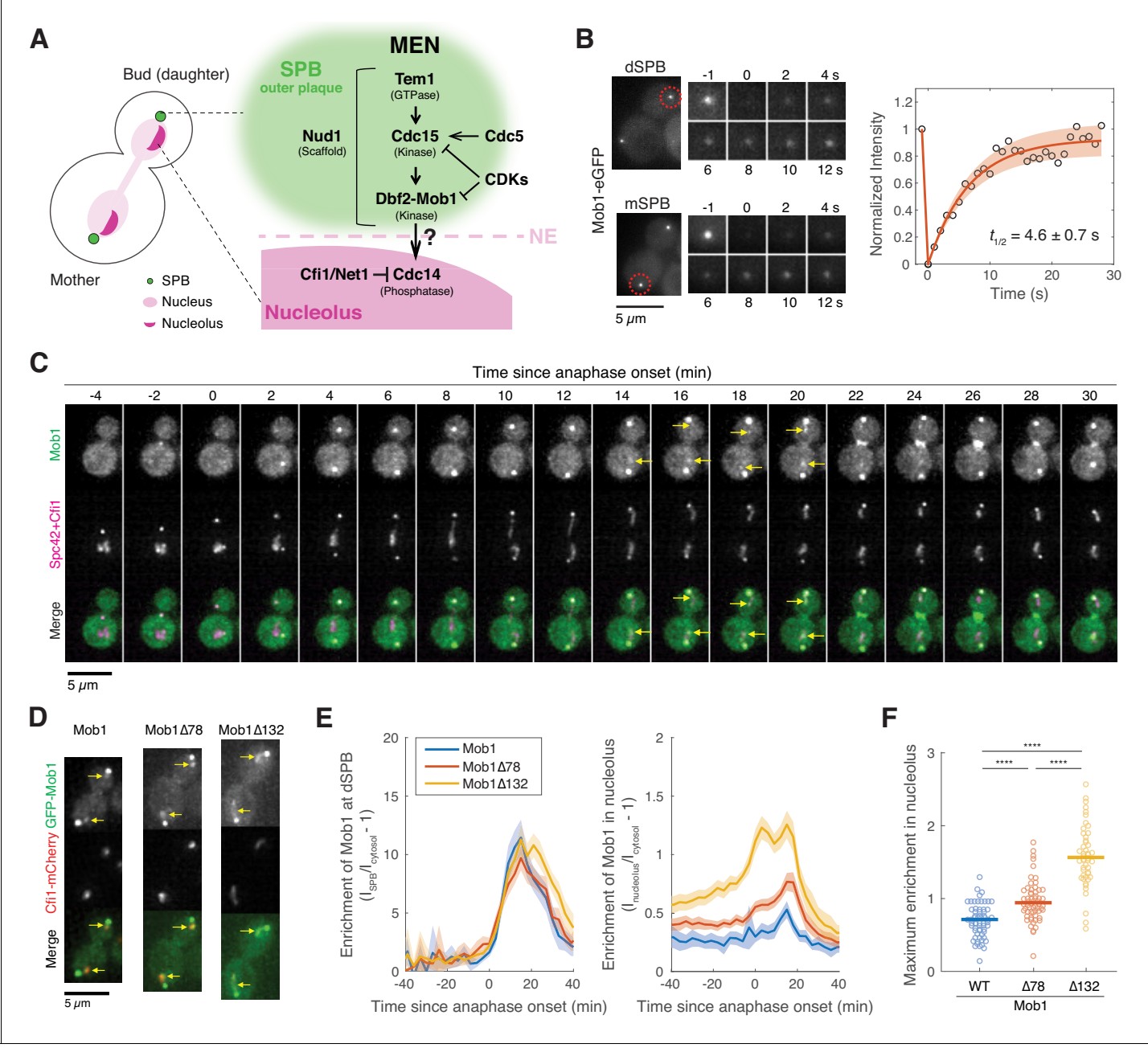

**Figure 1.** Dbf2-Mob1 transiently localizes to the nucleolus in late anaphase. (**A**) Major components of the mitotic exit network (MEN) and their subcellular localization. (**B**) Fluorescence recovery after photobleaching (FRAP) analysis of Mob1-eGFP (A39695). Red circles indicate the area of photo-bleaching. Cells were grown and imaged at room temperature in SC medium + 2% glucose. Graph to the right represents average measurements of double normalized fluorescence intensities (n = 6 cells) after correcting for photo-bleaching during acquisition. Red curve is the average fit and shaded area represents standard deviation (SD) of the fits. Half recovery time $t_{1/2}$ ± SD is indicated. (**C**) Localization of Mob1 during the cell cycle. A40257 (with Mob1-eGFP, Cfi1-mCherry and Spc42-mCherry) cells were grown at room temperature in SC medium + 2% glucose and imaged every minute for 2 hr. Arrows highlight the nucleolar localization. (**D**) Nucleolar localization of full-length (A39931) and N-terminally truncated (A39933 and A39935) Mob1. Cells were grown at room temperature in SC medium + 2% glucose and imaged every 3 min for 4 hr. Arrows highlight the nucleolar localization. (**E**) Enrichment of Mob1 (A41211, n = 62 cells), Mob1Δ78 (A41212, n = 60 cells), and Mob1Δ132 (A41213, n = 48 cells) at the daughter spindle pole body (dSPB; left) and in the nucleolus (right) as a function of cell cycle progression. Cells were grown at 25°C in SC medium + 2% glucose and imaged every 3 min for 4 hr. Single cell traces were aligned based on anaphase onset, as defined as spindle length >3 μm (*Figure 1—figure supplement 1F*, measured based on SPB marker Spc42-mCherry), and averaged. Solid lines represent the average, and shaded areas represent 95% confidence intervals. (**F**) Maximum enrichment of full-length Mob1 (WT) and truncated Mob1 (Mob1Δ78 and Mob1Δ132) in the nucleolus in anaphase of cells in (**E**). Solid lines represent the median. ****p<0.0001 by two-sided Wilcoxon rank sum test.

*Figure 1 continued on next page*

*Figure 1 continued*

The online version of this article includes the following figure supplement(s) for figure 1:

**Figure supplement 1.** N-terminally truncated Mob1 is hyperactive.

**Figure supplement 2.** Mob1's nucleolar localization correlates with mitotic exit network (MEN) activation and Cdc14 release from the nucleolus.

Tem1 activates its effector, the Hippo-like protein kinase Cdc15, presumably by recruiting Cdc15 to the SPBs (*Rock and Amon, 2011*; *Figure 1A*). Cdc15 then activates the LATS/NDR kinase Dbf2-Mob1 via a two-step process (*Rock et al., 2013*). Cdc15 first phosphorylates the MEN scaffold Nud1, a core component of the SPB outer plaque. This phosphorylation creates a docking site for Dbf2-Mob1 on Nud1, facilitating phosphorylation and activation of Dbf2-Mob1 by Cdc15 (*Mah et al., 2001*; *Rock et al., 2013*).

In addition to sensing spindle position, the MEN also integrates cues of cell cycle progression through the downstream kinases Cdc15 and Dbf2-Mob1 (*Campbell et al., 2019*). Two cell cycle events are sensed by the MEN: (1) Activity of the Polo-like kinase Cdc5: activation of Cdc15 depends on Cdc5 activity (*Rock and Amon, 2011*) which occurs only in mitosis (*Cheng et al., 1998*); (2) Initiation of anaphase: cyclin-dependent kinases (CDKs) phosphorylate Cdc15 and Mob1 thereby inhibiting their activity (*Jaspersen and Morgan, 2000*; *König et al., 2010*). At the onset of anaphase, CDK activity declines due to cyclin degradation, which lifts this inhibition. This loss of inhibition by CDKs creates a state whereby the MEN is poised for activation.

Once activated, the MEN promotes exit from mitosis by activating the phosphatase Cdc14. As an antagonist of CDKs, Cdc14 reverses CDK-dependent phosphorylation and promotes mitotic CDK inactivation thereby returning the cell to a G1 state (reviewed in *Stegmeier and Amon, 2004*). Given the central role of Cdc14 in promoting mitotic exit, it is not surprising that the phosphatase is tightly regulated. Cdc14 is sequestered in the nucleolus by its inhibitor Cfi1/Net1 from G1 until the onset of anaphase (*Shou et al., 1999*; *Visintin et al., 1999*). During anaphase, Cdc14 is released from its inhibitor and spreads throughout the nucleus and cytoplasm to dephosphorylate its targets.

Two pathways, the Cdc fourteen early anaphase release (FEAR) network and the MEN promote the dissociation of Cdc14 from its inhibitor during anaphase. Upon anaphase entry, Cdc14 is transiently released from the nucleolus by the FEAR network (reviewed in *Rock and Amon, 2009*). The FEAR network promotes dissociation of Cdc14 from Cfi1/Net1 by facilitating phosphorylation of Cfi1/Net1 by mitotic CDKs (*Azzam et al., 2004*; *Queralt et al., 2006*). This transient release, although not essential for exit from mitosis, is crucial for the timely execution of several anaphase events such as segregation of the nucleolus (*D'Amours et al., 2004*; *Sullivan et al., 2004*; *Torres-Rosell et al., 2004*) and MEN activation by counteracting CDK inhibition of MEN kinases (*Campbell et al., 2019*; *Jaspersen and Morgan, 2000*; *König et al., 2010*). In late anaphase, the activated MEN drives a more sustained and complete release of Cdc14 from the nucleolus which ultimately results in exit from mitosis. In the absence of MEN activity, Cdc14, after a transient FEAR network-mediated release from the nucleolus during early anaphase, is re-sequestered in the nucleolus and cells arrest in late anaphase.

Despite our extensive knowledge of the MEN and Cdc14, how the MEN promotes the sustained release of Cdc14 from its inhibitor in the nucleolus is not well understood. One contributing mechanism involves the phosphorylation of the nuclear localization signal (NLS) sequence in the C-terminus of Cdc14 by the MEN kinase Dbf2-Mob1 (*Mohl et al., 2009*). Inactivation of the NLS by the MEN promotes redistribution of Cdc14 via its nuclear export signal (NES) from the nucleus to the cytoplasm. However, retention of Cdc14 in the cytoplasm is not required for mitotic exit (*Bembenek et al., 2005*; *Kuilman et al., 2015*; *Mohl et al., 2009*). Furthermore, a Cdc14 mutant lacking the Dbf2 phosphorylation sites within its NLS is still released from the nucleolus in late anaphase (*Mohl et al., 2009*). These results suggest that MEN-mediated cytoplasmic retention of Cdc14 is not the main mechanism whereby the MEN activates Cdc14. Rather, the MEN must also disrupt the interaction between Cdc14 and its inhibitor Cfi1/Net1 in the nucleolus.

How does the MEN, activated at the outer plaque of SPBs in the cytosol, liberate Cdc14 from its inhibitor in the nucleolus? Proteome array screens for Dbf2-Mob1, the terminal kinase in the MEN, have identified Cfi1/Net1 as a potential Dbf2 substrate in vitro (*Mah et al., 2001*; *Ptacek et al., 2005*) suggesting that the MEN might liberate Cdc14 from Cfi1/Net1 through phosphorylating Cfi1/

Net1. However, given that Dbf2-Mob1 and Cfi1/Net1 normally reside in different cellular compartments, whether this phosphorylation occurs in vivo and contributes to Cdc14 activation are unclear. In addition, the Polo-like kinase Cdc5, an important regulator of mitotic exit and essential for Cdc14 activation, has also been shown to phosphorylate Cfi1/Net1 (*Loughrey Chen et al., 2002*; *Shou et al., 2002*; *Yoshida and Toh-e, 2002*). Whether and how this phosphorylation contributes to MEN-mediated Cdc14 release are not understood.

Here, we demonstrate that Dbf2-Mob1 serves as a molecular messenger traveling between the SPBs and the nucleolus to release Cdc14 from its inhibitor. We show that Dbf2-Mob1, normally kept out of the nucleus by Crm1, gains access to the nucleus following activation by Cdc15. We further demonstrate that Dbf2-Mob1 utilizes a nucleolar docking site created by Cdc5 to phosphorylate Cfi1/Net1, resulting in Cdc14 liberation. These findings define the molecular mechanisms of cross-compartment signal transmission in the MEN and provide a novel paradigm for how signaling can occur across organelle boundaries.

## Results

### Dbf2-Mob1 dynamically associates with SPBs

When the MEN is activated in anaphase, Dbf2-Mob1 is recruited to the outer plaque of SPBs by binding to Cdc15-phosphorylated Nud1 (*Rock et al., 2013*). However, immobilizing Dbf2-Mob1 at SPBs by fusing Mob1 to Nud1 disrupts MEN activity (*Rock et al., 2013*), suggesting that Dbf2-Mob1 is likely needed away from the SPBs for the MEN to function. Additionally, a small fraction of Dbf2-Mob1 was found to enter mitotic nuclei (*Stoepel et al., 2005*). Thus, we hypothesized that the MEN liberates Cdc14 from its nucleolar inhibitor through the dynamic shuttling of Dbf2-Mob1 between the outer plaque of the SPB and the nucleolus. We reasoned that as a messenger between the SPB and nucleolus, Dbf2-Mob1 needs to be mobile at the SPB. To test this hypothesis, we performed fluorescence recovery after photobleaching (FRAP) analysis on eGFP tagged Mob1 in anaphase cells (*Figure 1B*). We observed a rapid recovery of fluorescence with a half-recovery time of 4.6 ± 0.7 s (mean ± SD, $n$ = 6 cells) after photobleaching of Mob1-eGFP fluorescence either at the daughter (dSPB) or the mother (mSPB) SPB. This fast turnover rate (~1/500 of the total duration for Dbf2-Mob1's SPB localization in anaphase) indicates that localization of Dbf2-Mob1 to SPBs is highly dynamic.

### Dbf2-Mob1 transiently localizes to the nucleolus during anaphase

We next investigated whether Dbf2-Mob1 localizes to the nucleolus by live-cell fluorescence microscopy. Although subtle, we observed transient localization of Mob1-eGFP to the nucleolus in some cells as judged by co-localization with the nucleolar protein Cfi1/Net1. Importantly, this nucleolar localization was only observed in late anaphase cells after nucleolar segregation, when the MEN is normally active (*Figure 1C*).

It was reported previously that two N-terminally truncated Mob1 mutant proteins, Mob1Δ78 and Mob1Δ132 (Mob1 missing the first 78 and 132 amino acids, respectively), localize more prominently to the nucleus (*Stoepel et al., 2005*). We found that they also displayed increased nucleolar localization (*Figure 1D and F*). We hypothesize that the N-terminus of Mob1 harbors auto-inhibitory sequences that prevent access of the protein to the nucleolus. Hence, deleting these sequences ought to cause hyperactivation of Dbf2-Mob1. Indeed, we found that N-terminal truncation mutants of Mob1 partially suppressed temperature sensitive alleles of upstream MEN components (*cdc15-2* and *tem1-3*; *Figure 1—figure supplement 1A*). This suppression was not a result of elevated Mob1 protein levels because overexpression of Mob1 from the *GPD* promoter did not suppress the growth defect of *cdc15-2* or *tem1-3* mutants (*Figure 1—figure supplement 1A and B*), nor did it increase Mob1's nucleolar localization (*Figure 1—figure supplement 1C and D*). We conclude that N-terminal truncations result in enhanced nucleolar localization and hyperactivation of Mob1.

To further characterize the cellular localization of Dbf2-Mob1, we quantified the relative enrichment of full-length and truncated GFP-Mob1 at SPBs and in the nucleolus during the cell cycle (*Figure 1E*). Full-length Mob1 localized to SPBs and the nucleolus during anaphase. Localization of Mob1Δ78 and Mob1Δ132 to SPBs was similar to that of full-length Mob1. The nucleolar localization of Mob1 and truncated Mob1 (*Figure 1E*) correlated with MEN activation, as judged by Mob1

association with SPBs, translocation of the MEN activity reporter NLS$_{Cdc14}$ (*Campbell et al., 2019*) into the cytoplasm, and MEN-mediated Cdc14 release from the nucleolus (*Figure 1—figure supplement 2*). Consistent with earlier observations, the Mob1 truncations displayed significantly greater nucleolar enrichment relative to full-length Mob1 in anaphase (~30% and 120% increase on average for Mob1Δ78 and Mob1Δ132 respectively, *Figure 1F*). Mob1Δ78 localization to the nucleolus was, like full-length Mob1, restricted to anaphase but accumulated in the nucleolus to higher levels. In contrast, Mob1Δ132 displayed both greater and earlier nucleolar enrichment, evident already in metaphase. We conclude that Dbf2-Mob1 localizes to the nucleolus during anaphase when the MEN is active. N-terminal truncation mutants of Mob1 exhibit enhanced nucleolar localization and are hypermorphic. Given that the nucleolar localization of full-length Mob1 is quite subtle, we used the Mob1 truncation mutants as tools to study Dbf2-Mob1's nucleolar localization.

## Dbf2-Mob1 localizes to the nucleolus through interacting with Cfi1/Net1

To validate the nucleolar localization of Mob1 we observed by microscopy and to identify the potential nucleolar receptor for Dbf2-Mob1, we performed TurboID proximity-based labeling (*Branon et al., 2018*). We fused the promiscuous biotin ligase TurboID to the MEN components Mob1, Dbf2, Tem1, and Cdc15 and identified their protein interactors by streptavidin pull-down followed by mass spectrometry (MS) (*Figure 2—figure supplement 1A*). In this experiment, we identified the nucleolar protein Cfi1/Net1 as the top hit for Mob1- and Dbf2-TurboID labeling (*Figure 2A*, *Figure 2—figure supplement 1B*, *Supplementary file 3*). Biotinylation of Cfi1/Net1 by Mob1-TurboID was further confirmed by the detection of a biotinylated peptide of Cfi1/Net1 (*Figure 2—figure supplement 1C*). Importantly, Cfi1/Net1 was only detected in the labeling experiments where Mob1 or Dbf2 was tagged with TurboID but not when Tem1 or Cdc15 was used as baits (*Figure 2A*). In contrast, Nud1, the MEN scaffold protein at SPBs, was detected in the TurboID labeling experiments for all MEN proteins (*Figure 2A*).

We validated these MS findings using streptavidin gel-shift assays (*Fairhead and Howarth, 2015*; *Housley et al., 2014*). To monitor whether a target protein was biotinylated by the TurboID-tagged bait protein in vivo, we treated the denatured cell lysates with excess streptavidin prior to immunoblotting. Biotinylated form(s) of the target protein will migrate more slowly in SDS-PAGE due to binding of streptavidin, with each added biotin molecule causing a theoretical size increase of up to 53 kD, the size of a streptavidin tetramer (*Figure 2—figure supplement 1D*). Using this assay, we observed a slower migrating form of Nud1 in cell lysates from cells expressing all TurboID-tagged MEN proteins (*Figure 2B*). Slower migrating forms of Cfi1/Net1 were only observed in lysates obtained from cells expressing Mob1-TurboID (*Figure 2B*).

To determine whether Cfi1/Net1 was the sole receptor for Mob1 in the nucleolus, we characterized the localization of Mob1Δ78 and Mob1Δ132 in cells lacking *CFI1/NET1*. While still localized to SPBs, Mob1Δ78 and Mob1Δ132 no longer accumulated in the nucleolus during anaphase in *cfi1/net1Δ* cells (*Figure 2C and D*). Furthermore, when we overexpressed *CFI1/NET1* from the strong galactose-inducible *GAL1-10* promoter, nucleolar localization of both full-length and N-terminal truncation mutants of Mob1 was increased by at least 50% (full-length) and up to 300% (truncations) (*Figure 2E and F*). It is worth noting that the observed decrease in Mob1's dSPB localization with *CFI1/NET1* overexpression (*Figure 2F*) is likely the result of Cdc14 inhibition by Cfi1/Net1 (*Visintin et al., 1999*) leading to reduced MEN activation, similar to the decrease in Mob1's dSPB localization observed when Cdc14 activity is reduced by the *cdc14-3* mutation (*Campbell et al., 2019*). We conclude that Dbf2-Mob1 localization in the nucleolus during anaphase is mediated by interactions with Cfi1/Net1.

## Nucleolar localization of Dbf2-Mob1 depends on MEN activation

The localization pattern of Dbf2-Mob1 leads to the model in which Dbf2-Mob1 is activated by Cdc15 at SPBs. Active Dbf2-Mob1 then binds to and phosphorylates Cfi1/Net1, promoting the dissociation of Cdc14 from its inhibitor to carry out mitotic exit. This model predicts that the nucleolar localization of Dbf2-Mob1 depends on MEN activity. To test this prediction, we employed an analog-sensitive allele of *CDC15*, *cdc15-as1* (*Bishop et al., 2000*; *D'Aquino et al., 2005*). As expected, inhibition of *cdc15-as1* (through addition of the analog 1-NA-PP1) prevented localization of Mob1 to

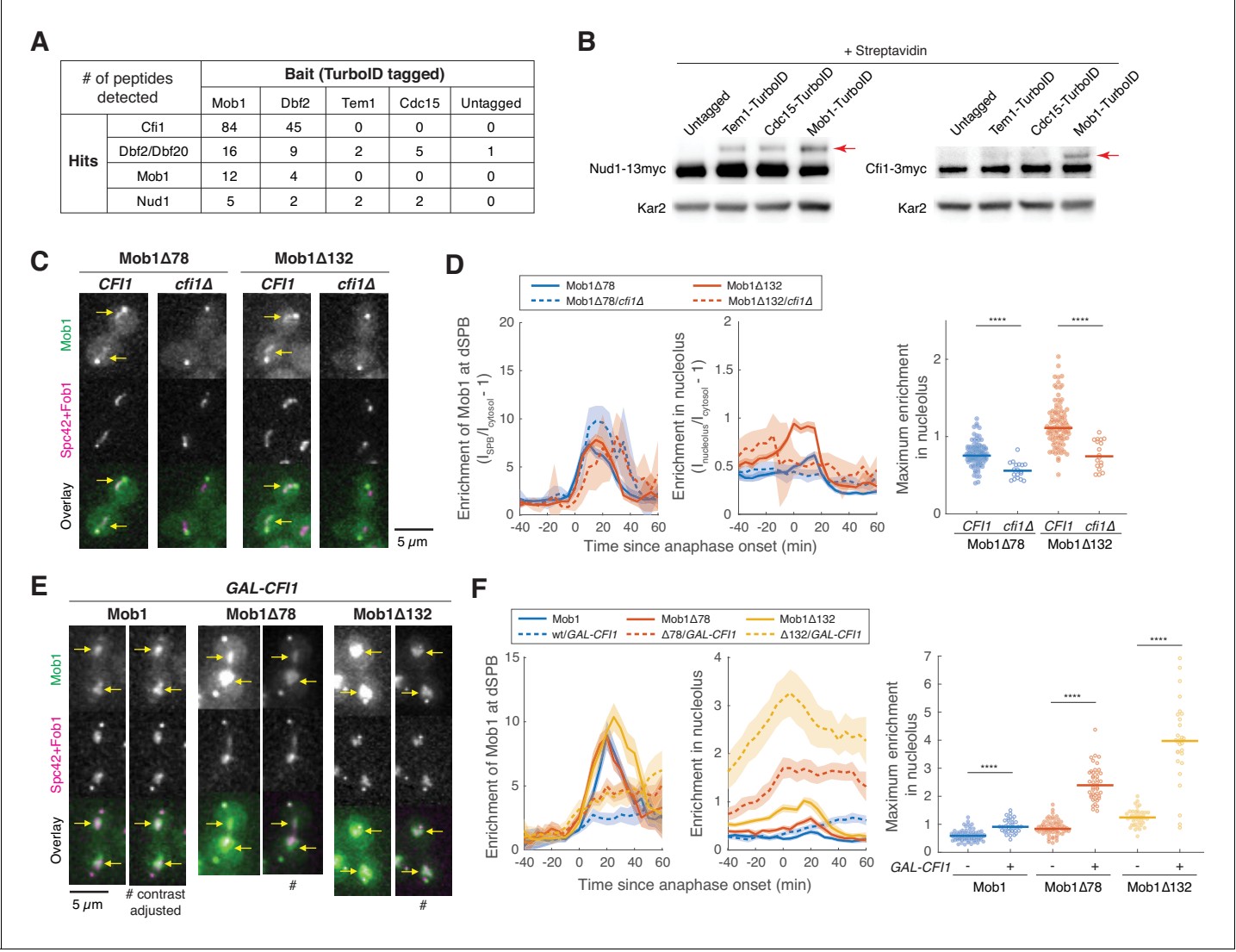

**Figure 2.** Dbf2-Mob1 localizes to the nucleolus through interacting with Cfi1/Net1. (**A**) Results of proximity-based biotinylation with TurboID for mitotic exit network (MEN) proteins (A41367, A41370, A41368, A41369, and A2588). Cells were grown at room temperature in YEP + 2% glucose + 50 μM biotin. (**B**) Streptavidin gel-shift assays to probe the interactions of TurboID-labeled MEN proteins with the MEN scaffold Nud1 (left, A11869, A41381, A41382 and A41380) or Cfi1/Net1 (right, A1638, A41406, A41407 and A41372). Cells were grown at room temperature in YEP + 2% glucose and lysates were treated with streptavidin and immunoblotted as indicated. Red arrows highlight biotinylated proteins. (**C and D**) Representative images (**C**) and quantification (**D**) of Mob1Δ78 localization in wild-type *CFI1/NET1* (A41344, n = 106 cells) or *cfi1/net1Δ* (A41347, n = 18 cells) cells and Mob1Δ132 localization in *CFI1/NET1* (A41345, n = 95 cells) or *cfi1/net1Δ* (A41348, n = 18 cells) cells. Cells were grown at 25°C in SC medium + 2% glucose and imaged every 5 min for 4 hr. Arrows highlight nucleolar localization. (**E and F**) Representative images (**E**) and quantification (**F**) of Mob1 localization in wild-type (A41343, n = 110 cells) or *GAL-CFI1/NET1* expressing cells (A41340, n = 71 cells), Mob1Δ78 localization in wild-type (A41344, n = 103 cells) or *GAL-CFI1/NET1* expressing cells (A41341, n = 68 cells), and Mob1Δ132 localization in wild-type (A41345, n = 71 cells) or cells expressing *GAL-CFI1/NET1* (A41342, n = 53 cells). # denotes that the image was linearly contrast adjusted to avoid over-saturation for Mob1Δ78 and Mob1Δ132. Cells were first grown at room temperature in SC medium + 2% raffinose. Cells were then mounted onto agarose pads made with SC medium + 1% raffinose + 1% galactose and imaged every 5 min for 5 hr at 25°C. Arrows highlight nucleolar localization. Solid lines represent the average of single cell traces aligned to anaphase onset while shaded areas represent 95% confidence intervals. For maximum enrichment, each dot represents a single cell. The solid lines represent the median. ****p<0.0001 by two-sided Wilcoxon rank sum test.

The online version of this article includes the following figure supplement(s) for figure 2:

**Figure supplement 1.** TurboID identified interaction between Dbf2-Mob1 and Cfi1/Net1.

SPBs and translocation of the MEN activity reporter NLS$_{Cdc14}$ into the cytoplasm (*Figure 3A*). Nucleolar localization of full-length Mob1 was also significantly reduced (*Figure 3A*). Nucleolar localization of the N-terminally truncated Mob1Δ78 and Mob1Δ132 mutants, on the other hand, was only moderately reduced (*Figure 3A*), which is consistent with the finding that these alleles partially suppress the temperature sensitive growth defect of *cdc15-2* cells (*Figure 1—figure supplement 1A*). We conclude that Mob1 localization to the nucleolus depends on MEN activity. The N-terminal hyperactive truncation mutations in Mob1 are less reliant on upstream MEN kinases for their nucleolar localization.

## Cdc15 regulates nuclear access of Dbf2-Mob1

How does Cdc15 cause Dbf2-Mob1 to localize to the nucleolus? To reach the nucleolus, Dbf2-Mob1 must first enter the nucleus. Considering that the size of the complex (102 kD) is above the passive diffusion limit of the nuclear envelope (~40–60 kD) (*Knockenhauer and Schwartz, 2016*), we hypothesized that nuclear access of Dbf2-Mob1 is regulated. We explored this possibility using the PhyB-PIF based light-inducible organelle targeting system (*Yang et al., 2013*; *Figure 3—figure supplement 1A*). We fused Mob1-eGFP to the PIF protein, which binds PhyB upon exposure to red light (650 nm). Using this system, we successfully recruited Mob1-eGFP-PIF to various subcellular locations such as SPBs or the outer mitochondrial membrane (*Figure 3—figure supplement 1B*).

To gauge Dbf2-Mob1's nuclear access in different cell cycle stages, we created a trap for nuclear Dbf2-Mob1 using nucleolar-anchored PhyB (PhyB-Sik1, *Figure 3B*). If Dbf2-Mob1 was shuttling between the nucleus and cytoplasm, PhyB-Sik1 would capture nuclear Mob1-eGFP-PIF when activated by light. Interestingly, in pre-anaphase cells, we were not able to capture a notable amount of nuclear Dbf2-Mob1 after 2 min of red-light activation. In contrast, in anaphase cells in which the MEN is active, Dbf2-Mob1 was readily recruited to the nucleolus within 2 min (*Figure 3C*). We obtained similar results with a Dbf2-PIF fusion (*Figure 3—figure supplement 1C*). Furthermore, when we activated PhyB every 15 min for 2 min to recruit Mob1 to the nucleolus in cells progressing through the cell cycle, Mob1 was only recruited to the nucleolus by light during anaphase when the protein was also present at SPBs (*Figure 3—figure supplement 1D*). By comparison, we were not able to recruit the upstream kinase Cdc15 to the nucleolus in any cell cycle stage (*Figure 3—figure supplement 1C*). Given that (1) Dbf2-Mob1 protein levels do not fluctuate considerably during the cell cycle (*Visintin and Amon, 2001*), that (2) the interaction between other PIF fusions and PhyB-Sik1 are not cell cycle regulated (*Yang et al., 2013*), and that (3) Mob1 can be recruited to cytoplasmic targets throughout the cell cycle (*Figure 3—figure supplement 1B*), we conclude that nuclear access of Dbf2-Mob1 is cell cycle regulated.

Because nuclear access of Dbf2-Mob1 correlates with MEN activation, we next tested whether it was regulated by the MEN by quantifying the relative enrichment of Mob1 in the nucleolus as a function of PhyB activation time (exposure to 650 nm light) in *CDC15* or *cdc15-2* cells. In cells with wild-type *CDC15*, light-induced nucleolar recruitment of Mob1 was higher in anaphase than pre-anaphase cells. In contrast, in *cdc15-2* cells this difference was abolished (*Figure 3D*). We conclude that in addition to activating Dbf2's kinase activity (*Mah et al., 2001*), Cdc15 regulates Dbf2-Mob1's nuclear access. Consistent with this notion, we find that Mob1Δ78, which partially suppresses the temperature sensitivity of the *cdc15-2* allele, displayed increased nuclear access in all cell cycle stages (*Figure 3—figure supplement 1E*).

## Dbf2-Mob1 is exported from the nucleus by Crm1

Dbf2-Mob1 is a substrate of the nuclear exportin Crm1 in vitro (*Kırlı et al., 2015*). To determine whether Crm1 plays a role in controlling Dbf2-Mob1 localization in vivo, we quantified the nucleolar localization of full-length and truncated Mob1 in cells carrying an allele of *CRM1* (*crm1T539C*) that is sensitive to the nuclear export inhibitor leptomycin B (LMB) (*Neville and Rosbash, 1999*). Treatment of *crm1T539C* cells with LMB led to an increase in nucleolar localization of both full-length and N-terminally truncated Mob1 (*Figure 3—figure supplement 2A*), suggesting that Crm1 controls nuclear export of Dbf2-Mob1.

Crm1 recognizes substrates with a leucine-rich NES. To test whether there was a functional NES in Dbf2 or Mob1, we overexpressed Dbf2 or Mob1 from the galactose-inducible *GAL1-10* promotor. Overexpressed Mob1 was enriched in the nucleus (*Figure 3—figure supplement 2B*) similar to what

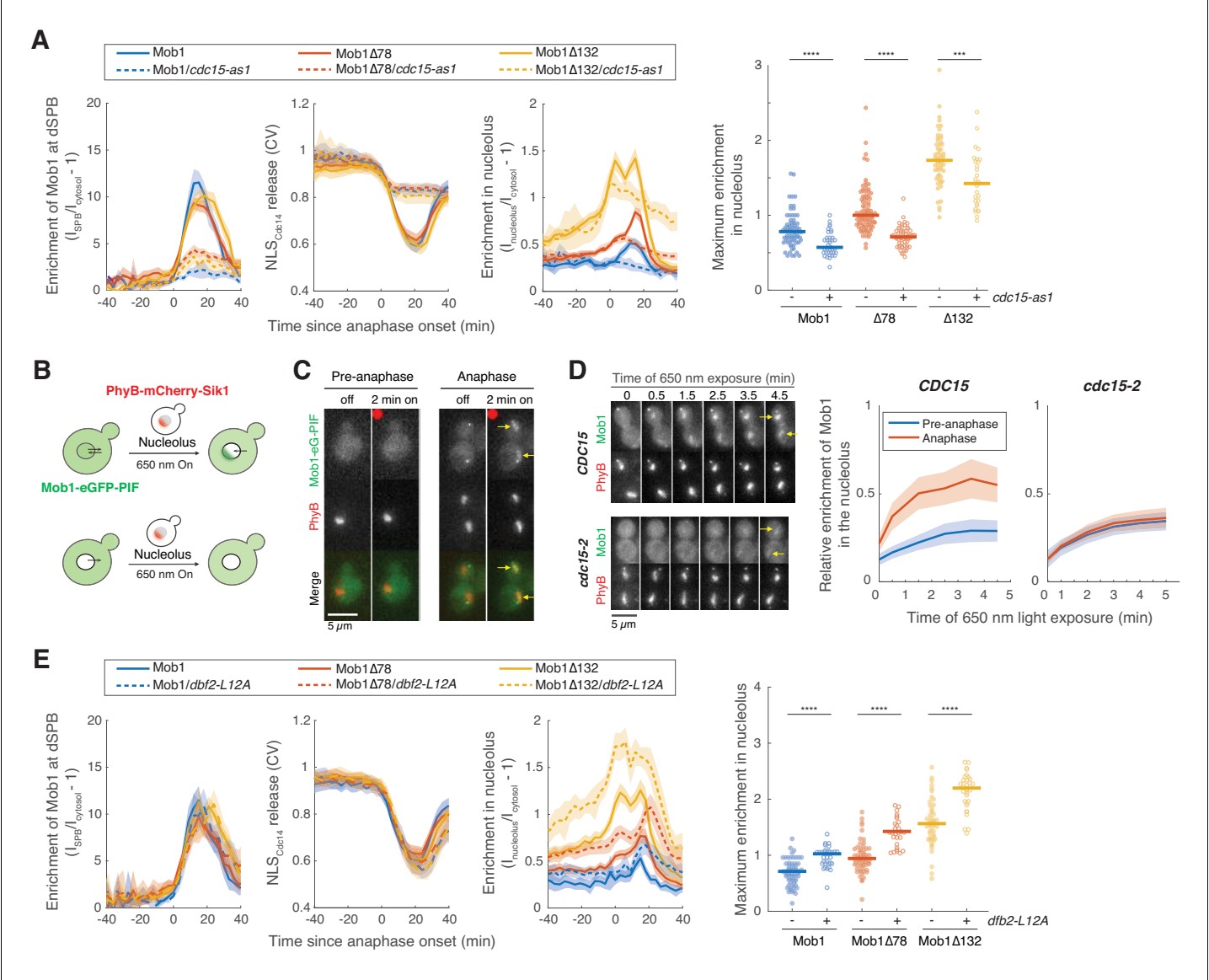

**Figure 3.** *CDC15* regulates nuclear access of Dbf2-Mob1. (**A**) Enrichment of Mob1 at the daughter spindle pole body (dSPB), in the nucleolus, and Dbf2-Mob1's kinase activity were determined in cells going through anaphase in *CDC15* (A41211, A41212, and A41213; n = 74, 94, and 55 cells respectively) or *cdc15-as1* (A41214, A41215, and A41216; n = 37, 63, and 30 cells respectively) cells. Cells were grown at 25°C in SC medium + 2% glucose and 10 μM 1-NA-PP1 and imaged every 3 min for 4 hr. (**B**) Probing Dbf2-Mob1's nuclear access by recruiting Mob1 to the nucleolus with the PhyB-PIF optogenetics system. By anchoring PhyB to the nucleolus, diffuse nuclear Dbf2-Mob1, if present, can be visualized by recruiting Mob1-eGFP-PIF to the nucleolus. (**C**) Recruiting Mob1 to the nucleolus at different cell cycle stages. A40260 cells were grown at 25°C in SC medium + 2% glucose, imaged after a 2 hr incubation with 31.25 μM PCB in the dark. Red dot denotes the frame where 650 nm light was applied to activate PhyB. Yellow arrows highlight the light-induced recruitment. (**D**) Recruitment of Mob1 to the nucleolus in *CDC15* (A41360) or *cdc15-2* (A41361) cells. Quantifications of Mob1's enrichment in the nucleolus as a function of PhyB activation time in *CDC15* (A41360, n = 27 and 16 cells for pre-anaphase and anaphase respectively) or *cdc15-2* (A41361, n = 14 and 36 cells for pre-anaphase and anaphase respectively) cells. Cells were grown at room temperature in SC medium + 2% glucose, incubated with 12.5 μM PCB for 2 hr in the dark, and shifted to 34°C for 50 min before imaging. (**E**) Nucleolar enrichment of full-length and truncated Mob1 in wild-type *DBF2* or *dbf2-L12A* cells (A41394, A41395, and A41396; n = 32, 28, and 31 cells respectively). Wild-type traces for comparison were the same as in *Figure 1E*. Cells were grown similarly as in *Figure 1E*. For graphs in (**A**) and (**E**), solid lines represent the average of single cell traces aligned to anaphase onset. Shaded areas represent 95% confidence intervals. For maximum enrichment, each dot represents a single cell. Solid lines represent the median. ****p<0.0001; ***p<0.001 by two-sided Wilcoxon rank sum test.

The online version of this article includes the following figure supplement(s) for figure 3:

**Figure supplement 1.** Examining nuclear access of Dbf2-Mob1 by optogenetics.

**Figure supplement 2.** Identification of a functional nuclear export signal (NES) in Dbf2.

**Figure supplement 3.** Phosphorylation of Dbf2's nuclear export signal (NES) partially regulates Dbf2-Mob1's nuclear access.

we observed for *pGPD-GFP-MOB1* (*Figure 1—figure supplement 1C*). In contrast, when we overexpressed Mob1 together with Dbf2, Mob1 was no longer nuclear enriched, suggesting that Dbf2 not Mob1 harbors a NES. Consistently, when overexpressed on its own, Dbf2 exhibited diffuse localization but inhibition of *crm1T539C* with LMB led to accumulation of Dbf2 in the nucleus (*Figure 3—figure supplement 2C*).

Sequence analysis identified a putative NES sequence in the N-terminus of Dbf2 beginning with L12 (*Figure 3—figure supplement 2D*). Incidentally, we noticed that this leucine was mutated to methionine in a previously isolated hyperactive allele of *DBF2* (*DBF2-HyA*)(*Geymonat et al., 2009*). Dbf2-HyA, when overexpressed, was nuclear enriched (*Figure 3—figure supplement 2E*). Inspired by this observation, we overexpressed *DBF2* mutants where L12 had been mutated to either methionine (*dbf2-L12M*) or alanine (*dbf2-L12A*). Both mutants accumulated in the nucleus (*Figure 3—figure supplement 2E*). Furthermore, we found that the first 23 amino acids of Dbf2 were sufficient to drive nuclear export of eGFP (*Figure 3—figure supplement 2E,G*). To test whether the NES in Dbf2 influences Dbf2-Mob1's nucleolar localization in anaphase under normal expression level, we characterized Mob1's cellular localization in *dbf2-L12A* and observed an increase in nucleolar localization by at least 40% for both full-length and N-terminally truncated Mob1 (*Figure 3E*).

A previous phosphoproteomic study reported that S17 and S20 within the NES of Dbf2 are phosphorylated in anaphase-arrested cells (*Holt et al., 2009*). We found that mutating S17 and S20 to phospho-mimetic residues (S17,20D or S17,20E) disrupted the NES whereas mutating these residues to alanine (S17,20A) retained the NES activity of Dbf2 (*Figure 3—figure supplement 2F and G*). We propose that phosphorylation of S17 and S20 is regulated, possibly by Cdc15, to control nuclear access of Dbf2-Mob1. Nucleolar localization of Mob1, particularly of Mob1Δ78, was reduced in cells harboring the *dbf2-S17,20A* allele compared to cells with wild-type *DBF2* (*Figure 3—figure supplement 3A*). However, cells carrying the *dbf2-S17,20A* allele, while exhibiting reduced nuclear access of Mob1 during all cell cycle stages, still showed differential nuclear access between pre-anaphase and anaphase, as is observed in *DBF2* cells (*Figure 3—figure supplement 3B*). This observation suggests that additional regulatory mechanism(s) control Dbf2-Mob1's nuclear access. In contrast, cells harboring the *dbf2-S17,20D* allele exhibited increased nuclear access of Mob1 during all cell cycle stages (*Figure 3—figure supplement 3B*), confirming that Dbf2-Mob1 is normally kept out of the nucleus through Dbf2's NES. The NES sequence in Dbf2 is well conserved among Saccharomycetes (*Figure 3—figure supplement 2D*) suggesting that regulated nuclear access of Dbf2-Mob1 is conserved at least across this class of fungi.

## Nucleolar localization of Dbf2-Mob1 is regulated by Cdc5

The analysis of nuclear access and nucleolar localization of the N-terminal truncations of Mob1 indicated that the MEN is not the only pathway controlling Dbf2-Mob1's nucleolar localization. The truncation mutants localize to the nucleolus in a manner largely independent of the MEN, yet their nucleolar localization is still restricted to metaphase and anaphase (*Figure 3A*). This restriction of nucleolar localization is not due to limited nuclear access. Truncated Mob1 mutants have increased nuclear access prior to anaphase (*Figure 3—figure supplement 1E*). These data indicate that nucleolar localization or the interaction between Dbf2-Mob1 with Cfi1/Net1 is regulated by additional factors.

An obvious candidate for this additional regulator is the Polo-like kinase Cdc5, which is active throughout mitosis and plays multiple essential roles in mitotic exit (*Lee et al., 2005*). As part of both the FEAR network (*Rock and Amon, 2009*; *Stegmeier et al., 2002*) and the MEN, Cdc5 is indispensable for Cdc14's nucleolar release. However, the exact role(s) of Cdc5 during this process is not fully understood. To determine whether Cdc5 regulates binding of Dbf2-Mob1 to Cfi1/Net1, we examined the consequences of inhibiting Cdc5's kinase activity on nucleolar localization of Dbf2-Mob1 using an analog-sensitive allele of *CDC5* (*cdc5-as1*). Consistent with the known functions of Cdc5 in MEN activation, we observed loss of Mob1's SPB localization and Dbf2-Mob1's kinase activity as monitored by translocation of the $NLS_{Cdc14}$ reporter into the cytoplasm when Cdc5 was inhibited (*Figure 4A*). Nucleolar localization of Mob1, Mob1Δ78, and Mob1Δ132 was also lost in cells lacking Cdc5 activity (*Figure 4A*). This is in direct contrast to Cdc15 inhibition, where the nucleolar localization of N-terminal Mob1 truncation mutants particularly Mob1Δ132 was only partially reduced (*Figure 3A*). These results suggested that Cdc5 regulates Dbf2-Mob1's nucleolar localization independently of its role in activating Cdc15.

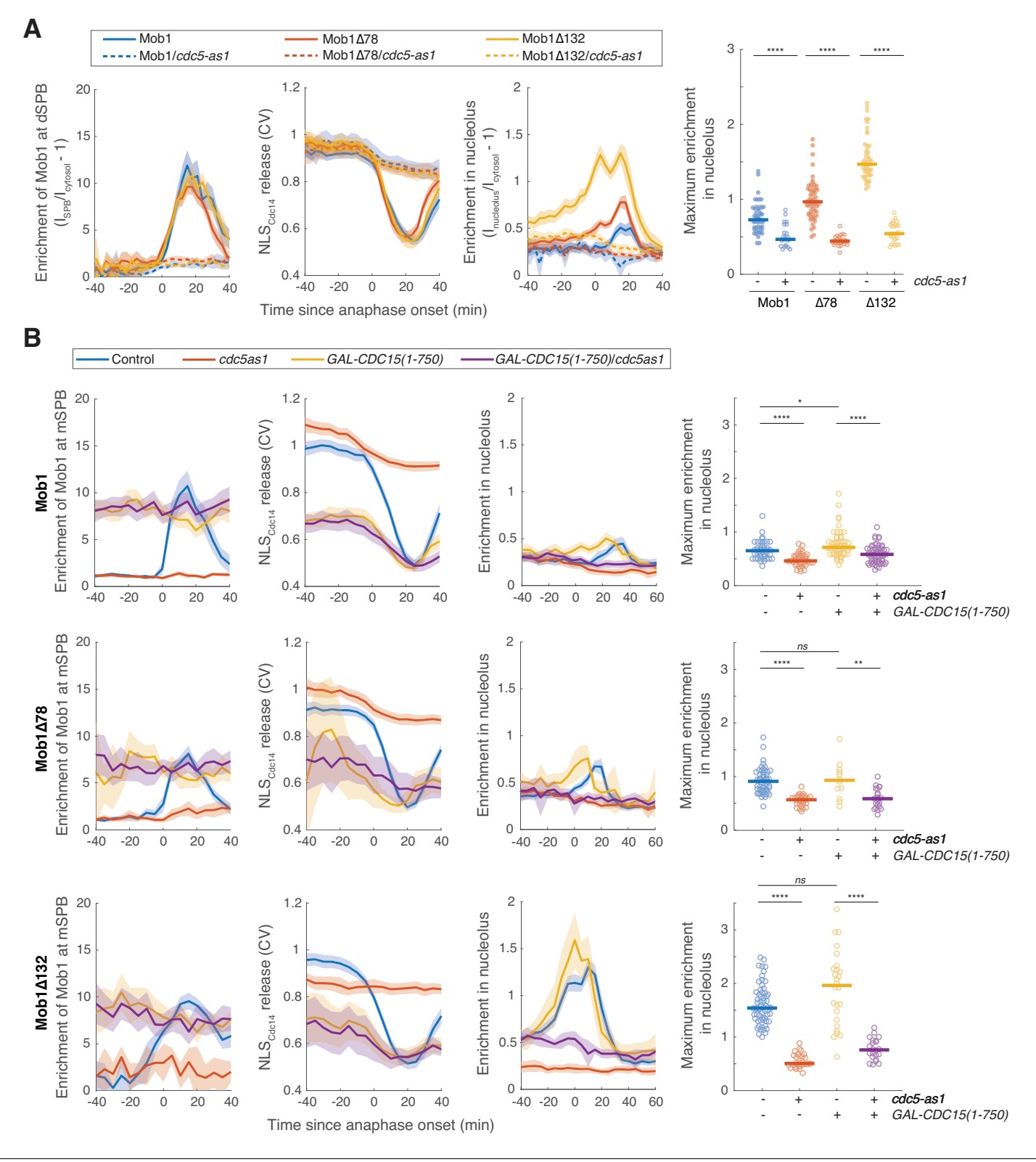

**Figure 4.** Nucleolar localization of Dbf2-Mob1 is regulated by *CDC5* independently of *CDC15*. (**A**) Enrichment of Mob1 at the daughter spindle pole body (dSPB), in the nucleolus, and Dbf2-Mob1's kinase activity in cells wild type for *CDC5* (A41211, A41212, and A41213; *n* = 49, 60, and 47 cells respectively) or harboring a *cdc5-as1* allele (A41334, A41335, and A41336; *n* = 23, 30, and 28 cells respectively). Cells were grown at 25°C in SC medium + 2% glucose and 5 μM CMK and imaged every 3 min for 4 hr. (**B**) Cells harboring *GAL-CDC15(1-750)* and *cdc5-as1* either containing *eGFP-MOB1*

*Figure 4 continued on next page*

Figure 4 continued

(A41211, A41334, A41376, and A41337; $n = 44, 41, 58$, and 61 cells respectively), or *eGFP-MOB1Δ78* (A41212, A41335, A41377, and A41338; $n = 54, 30$, 12, and 22 cells respectively), or *eGFP-MOB1Δ132* (A41213, A41336, A41378, and A41339; $n = 62, 28, 26$, and 22 cells respectively) were analyzed to determine Mob1 localization. Localization to the mother SPB (mSPB) instead of dSPB was quantified here because cells expressing *GAL-CDC15(1-750)* often exit from mitosis in the mother (without movement of a SPB into the bud). For cells exited with two SPBs in the mother cell, maximum intensities of the two SPBs were used. Cells were grown at 25°C in SC medium + 1% raffinose, 1% galactose, and 5 µM CMK and imaged every 5 min for 5 hr. Solid lines represent the average of single cell traces aligned to anaphase onset. Shaded areas represent 95% confidence intervals. For maximum enrichment, each dot represents a single cell. Solid lines represent the median. ****$p<0.0001$; **$p<0.01$; *$p<0.05$ by two-sided Wilcoxon rank sum test. The online version of this article includes the following figure supplement(s) for figure 4:

**Figure supplement 1.** Nucleolar localization of Dbf2-Mob1 does not depend on the Cdc fourteen early anaphase release (FEAR) network.

To directly determine whether *CDC5* regulated Dbf2-Mob1's nucleolar localization independently of the MEN, we took advantage of a hyperactive *CDC15* allele, *GAL-CDC15(1-750)* (*Bardin et al., 2003*), which is active even in the absence of *CDC5* (*Rock and Amon, 2011*). When *GAL-CDC15(1-750)* was expressed, Mob1's SPB localization and Dbf2-Mob1's kinase activity was no longer restricted to anaphase but rather was high throughout the cell cycle as a result of MEN hyper-activation (*Figure 4B*). Interestingly, *GAL-CDC15(1-750)* did not abolish cell-cycle regulation of Mob1's nucleolar localization but rather advanced it to early anaphase and metaphase (*Figure 4B*). In cells expressing *GAL-CDC15(1-750)*, inactivation of *CDC5* still abolished nucleolar localization of both full-length and the hyperactive N-terminally truncated Mob1 while Mob1 binding to SPBs was unaffected (*Figure 4B*). These results demonstrate that nucleolar localization of Dbf2-Mob1 directly depends on Cdc5 independently of its role in MEN activation.

Could *CDC5* regulate the nucleolar localization of Dbf2-Mob1 through its role in the FEAR network? To test this, we quantified Mob1's nucleolar localization in cells lacking the FEAR network component *SLK19* (*slk19Δ*). MEN activation (as determined by Mob1 localization to the dSPB and nuclear release of the NLS$_{Cdc14}$ reporter) and as a result mitotic exit were considerably delayed and more variable in *slk19Δ* cells (*Figure 4—figure supplement 1A*). Consistent with a delay in MEN activation, nucleolar localization of Mob1 and Mob1Δ78 but not Mob1Δ132 was also delayed. Importantly, maximum enrichment of Mob1 in the nucleolus was not reduced in *slk19Δ* cells for all three forms of Mob1 (*Figure 4—figure supplement 1B*). We conclude that Cdc5 regulates Dbf2-Mob1's nucleolar localization through mechanisms in addition to its role in the MEN and the FEAR network.

## Cdc5 and Dbf2-Mob1 phosphorylate Cfi1/Net1 at distinct sites

Our results suggest a model where Cdc5 promotes the interaction between Dbf2-Mob1 and its nucleolar receptor Cfi1/Net1, likely through phosphorylating Cfi1/Net1. This interaction then facilitates phosphorylation of Cfi1/Net1 by Dbf2-Mob1 to bring about the release of Cdc14 from Cfi1/Net1. Cfi1/Net1 is a highly phosphorylated protein with 64 known phosphorylation sites in vivo (*Holt et al., 2009*; *Swaney et al., 2013*). About one fifth of these sites were identified as CDK targets (*Holt et al., 2009*) including six key CDK sites whose phosphorylation is controlled by the FEAR network (*Azzam et al., 2004*). To map sites in Cfi1/Net1 that are phosphorylated in a *CDC5* or MEN-dependent manner, we performed phosphoproteomic analyses on wild-type anaphase cells and cells in which Cdc5 or Cdc15 were inhibited using the *cdc5-as1* and *cdc15-as1* alleles, respectively (*Figure 5—figure supplement 1A*). This analysis identified 44 of the 64 previously known sites in Cfi1/Net1 and 18 new sites (*Supplementary file 4*). To achieve complete or close to complete coverage of Cfi1/Net1's phosphorylation sites in anaphase, we also performed immunoprecipitation-mass spectrometry (IP-MS) for Cfi1/Net1 in anaphase enriched cultures and identified nine additional sites (*Supplementary file 4*) resulting in an astonishing total of 91 phosphorylation sites in Cfi1/Net1. These phosphorylation sites appear to cluster in regions of disorder as predicted by the PONDR score (*Romero et al., 1997*; *Figure 5A*).

By comparing the peptide signals between wild-type, *cdc15-as1*, and *cdc5-as1* cells in our quantitative phosphoproteomics data set, we identified phosphorylation sites that depended on Cdc15 or Cdc5 activity or both (*Figure 5—figure supplement 1B–E*). Among them, we found 11 *CDC15*-dependent and 22 *CDC5*-dependent sites in Cfi1/Net1. Six of the *CDC15*-dependent sites fit Dbf2-Mob1's preferred sequence motif RXXS* (* represents the phosphorylation site) (*Mah et al., 2005*), supporting our model that Dbf2-Mob1 phosphorylates Cfi1/Net1. Given that Cdc5 activates Cdc15,

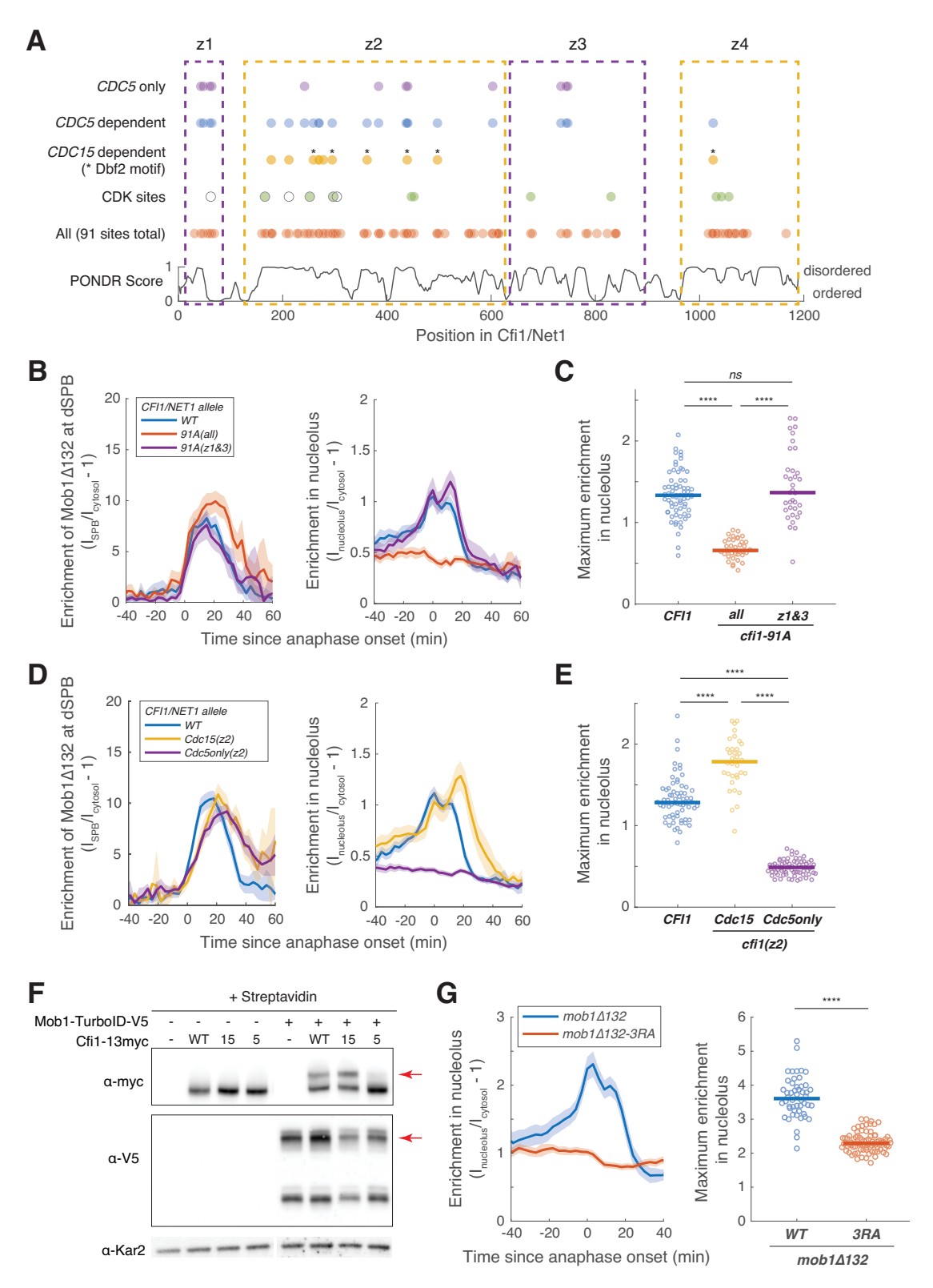

**Figure 5.** Cdc5 promotes Dbf2-Mob1's nucleolar localization by phosphorylating Cfi1/Net1. (**A**) Distribution of all, CDK sites, *CDC15*- and *CDC5*-dependent phosphorylation sites (***Supplementary file 4***) and disordered regions in Cfi1/Net1. For CDK sites, open circles represent sites identified and mutated in ***Azzam et al., 2004*** (*cfi1/net1-6Cdk*) and filled circles represent sites identified in ***Holt et al., 2009***. Dashed boxes denote the four zones. (**B–E**) Localization of Mob1Δ132 in *CFI1/NET1* (A41411, *n* = 67 cells for **B and C** and 66 cells for **D and E**), *cfi1-91A* mutants (A41412 and A41413, *n* = 36

*Figure 5 continued on next page*

Figure 5 continued

and 35 cells), *cfi1-Cdc15(z2)* (A41593, *n* = 34 cells) or *cfi1-Cdc5only(z2)* (A41594, *n* = 69 cells). Cells were grown at 25°C in SC medium + 2% glucose and imaged every 3 min for 4 hr. (**F**) Streptavidin gel-shift assays to probe the interactions between TurboID-tagged Mob1 and different *CFI1/NET1* alleles (from left to right: A2587, A41596, A41597, A41598, A41379, A41611, A41612, A41613). -, not tagged; WT, wild-type Cfi1-13myc; 15, Cfi1-Cdc15(z2)—13myc; 5, Cfi1-Cdc5only(z2)—13myc. Cells were grown at room temperature in YEP + 2% glucose and lysates were treated with streptavidin and immunoblotted as indicated. Red arrows highlight biotinylated proteins. (**G**) Enrichment of Mob1Δ132 (A41664, *n* = 50 cells) or Mob1Δ132-3RA (A41665, *n* = 85 cells) in the nucleolus. Cells were grown at 25°C in SC medium + 2% glucose and imaged every 3 min for 4 hr. For graphs in (**B–E and G**), solid lines represent the average of single cell traces aligned to anaphase onset. Shaded areas represent 95% confidence intervals. For maximum enrichment, each dot represents a single cell. Solid lines represent the median. ****$p<0.0001$ by two-sided Wilcoxon rank sum test.

The online version of this article includes the following figure supplement(s) for figure 5:

**Figure supplement 1.** Phosphoproteomics identifies *CDC15* (mitotic exit network [MEN]) and *CDC5* dependent phosphorylation in anaphase.
**Figure supplement 2.** Phosphorylation of Cfi1/Net1 modulates Dbf2-Mob1's nucleolar localization.
**Figure supplement 3.** Identification of *CDC5*-dependent phosphorylation sites in metaphase.
**Figure supplement 4.** Cdc5 interacts with Cfi1/Net1 in vivo.
**Figure supplement 5.** Dbf2-Mob1 localizes to the nucleolus via Mob1's phosphoserine-threonine binding domain.

sites that depended on *CDC15* ought to also depend on *CDC5*. This was indeed the case for 10 of the 11 *CDC15*-dependent phosphorylation sites in Cfi1/Net1.

To identify sites that only depended on *CDC5* but not *CDC15*, we subtracted *CDC15*-dependent sites from *CDC5*-dependent sites yielding 12 sites (denoted as *CDC5*-only, *Figure 5A*). Based on the distribution of all 91 phosphorylation sites and degree of disorder, we divided Cfi1/Net1 into four zones: residues 31–69 (z1, seven sites), 160–615 (z2, 53 sites), 676–840 (z3, 12 sites), and 1017–1166 (z4, 19 sites) (*Figure 5A*). *CDC15*-dependent phosphorylation sites were concentrated in zone 2, whereas *CDC5*-dependent sites were also found in zones 1 and 3. In contrast to Cfi1/Net1, all the *CDC5*-dependent phosphorylation sites in Dbf2 and Mob1 were also *CDC15*-dependent (*Figure 5—figure supplement 1F*). These data indicate that Cdc5 and Dbf2-Mob1 directly phosphorylate Cfi1/Net1. Dbf2-Mob1 on the other hand is a direct substrate of Cdc15 but not Cdc5.

## Cdc5 promotes Dbf2-Mob1's nucleolar localization by phosphorylating Cfi1/Net1

Having identified phosphorylation sites within Cfi1/Net1, we next asked whether they were important for the interaction between Cfi1/Net1 and Dbf2-Mob1. We generated a *CFI1* allele in which all 91 phosphorylation sites were mutated to alanine (*cfi1-91A*). Cells harboring this allele as the sole source of *CFI1/NET1* were viable and progressed through anaphase with only a slight delay in mitotic exit as judged by the timing of Mob1's dissociation from the SPBs (*Figure 5B*). Interestingly, Mob1Δ132, which showed the most pronounced nucleolar localization among all Mob1 alleles analyzed, still localized to the SPBs during anaphase in *cfi1-91A* cells, but failed to accumulate in the nucleolus (*Figure 5B and C*, *Figure 5—figure supplement 2A*). We conclude that Cfi1/Net1 phosphorylation is required for interacting with Dbf2-Mob1.

Next, we tested whether *CDC5*-dependent phosphorylation of Cfi1/Net1 regulated Dbf2-Mob1's nucleolar localization. We focused our analysis on zone 2 of Cfi1/Net1 because previous studies had shown that the first 621 amino acids of the protein are sufficient to confer Cdc14 regulation (*Azzam et al., 2004*). We further note that mutating phosphorylation sites in zones 1 and 3 of Cfi1/Net1 did not affect the nucleolar localization of Mob1Δ132 (*Figure 5B and C*, *Figure 5—figure supplement 2A*). We generated a *CFI1/NET1* allele with mutated phosphorylation sites in zone 2 that were phosphorylated in a *CDC5*-dependent but *CDC15*-independent manner (henceforth *cfi1-Cdc5only(z2)*). For comparison we generated a *CFI1/NET1* allele in which we only mutated sites that were phosphorylated in a *CDC15*-dependent manner (henceforth *cfi1-Cdc15(z2)*). Analysis of Mob1Δ132 localization in these mutants revealed that, similar to inhibition of Cdc5, *cfi1-Cdc5only(z2)* abolished the nucleolar localization of Mob1Δ132, while *cfi1-Cdc15(z2)* slightly increased nucleolar localization of the protein (*Figure 5D and E*, *Figure 5—figure supplement 2B*). To validate these findings, we performed TurboID labeling experiments followed by streptavidin gel-shift assays to probe the interaction between TurboID tagged full-length Mob1 and 13Myc tagged Cfi1/Net1. The slower migrating form corresponding to biotinylated Cfi1/Net1 was absent in cells expressing *cfi1-*

*Cdc5only(z2)* but present in cells expressing *CFI* or *cfi1-Cdc15(z2)* (*Figure 5F*). We conclude that phosphorylation of Cfi1/Net1 by Cdc5 is required for Dbf2-Mob1 binding to the protein.

Three lines of evidence indicate that Cdc5 directly phosphorylates Cfi1/Net1. First, consistent with the finding that Cdc5 is already active in metaphase, 10 out of 12 *CDC5*-only sites identified in anaphase cells are already phosphorylated in cells arrested in metaphase using the microtubule depolymerizing drug nocodazole (*Figure 5—figure supplement 3*) with six of them also determined to be *CDC5*-dependent in metaphase (*Figure 5—figure supplement 3E*). Second, we found considerable overlap between *CDC5*-dependent phosphorylation sites in vivo and sites identified in vitro (*Loughrey Chen et al., 2002*; *Shou et al., 2002*). Four out of five in vivo Cdc5only sites in region 1–341 were previously found to be phosphorylated by Cdc5 in vitro (*Supplementary file 4*). Third, Cdc5 and Cfi1/Net1 interact with each other in vivo as determined by TurboID labeling (*Figure 5—figure supplement 4A*). The biotinylation of Cfi1/Net1 by Cdc5-TurboID was further confirmed by the detection of biotinylated peptides in Cfi1/Net1 as well as the streptavidin gel-shift assay (*Figure 5—figure supplement 4B–C*). We conclude that Cdc5 phosphorylates Cfi1/Net1 at the onset of metaphase which serves as a priming event for Dbf2-Mob1 binding to Cfi1/Net1 in anaphase. We note that these findings also explain why Mob1Δ132's nucleolar localization is already evident in metaphase (*Figure 4B*).

How does phosphorylation of Cfi1/Net1 by Cdc5 promote its binding to Dbf2-Mob1? Dbf2-Mob1 binds to Cdc15-phosphorylated Nud1 through Mob1's phosphoserine-threonine binding domain (*Rock et al., 2013*). We propose that a similar mechanism might also mediate the interaction between Dbf2-Mob1 and Cdc5-phosphorylated Cfi1/Net1. To test this possibility, we introduced the mutation R253A,R254A,R257A (3RA), shown to abolish Mob1's ability to bind phosphopeptides (*Rock et al., 2013*), to Mob1Δ132 and compared the localization of Mob1Δ132-eGFP and Mob1Δ132-3RA-eGFP expressed ectopically (*Figure 5—figure supplement 5A*). The 3RA mutation, while retaining nuclear access, abolished both the SPB and nucleolar localization of Mob1Δ132 (*Figure 5G* and *Figure 5—figure supplement 5*). These data strongly support a model in which Dbf2-Mob1 binds to Cdc5-phosphorylated Cfi1/Net1 via Mob1's phosphoserine-threonine binding domain.

## Dbf2-Mob1 promotes the release of Cdc14 from the nucleolus through Cdc5-mediated priming of Cfi1/Net1

To determine whether the *CDC5* and MEN-dependent phosphorylation sites in Cfi1/Net1 regulate the interaction between Cfi1/Net1 and Cdc14 as our model predicted, we examined the consequences of disrupting these phosphorylation sites on the release of Cdc14 from the nucleolus. We first determined which region of Cfi1/Net1 was mediating phospho-regulation of this interaction. We mutated all phosphorylation sites in the individual zones as well as in combination and analyzed the effects on Cdc14 release from the nucleolus. This analysis revealed that only phosphorylation in zone 2 controlled Cdc14 release from Cfi1/Net1 (*Figure 6—figure supplement 1A–B*). It is important to note that mutating the phosphorylation sites in zone 2 also affected the ability of Cfi1/Net1 to bind Cdc14, as judged by the lower degree of Cdc14 nucleolar sequestration prior to anaphase and localization of Cdc14 to the dSPB prior to anaphase (*Figure 6—figure supplement 1C*). This finding indicates that the same residues involved in regulating the interaction between Cfi1/Net1 and Cdc14 are also important for forming the complex in the first place and mutating them to alanine weakens this interaction. Alternatively, mutating so many residues at once (53 sites in zone 2) could change the structure of Cfi1/Net1 and thus disrupt binding to Cdc14. Nevertheless, the increased level of free Cdc14 in the cell with *cfi1-91A* and the zone 2-phosphomutant explains why such severe defect in Cdc14 release from the nucleolus did not cause a significant anaphase delay as assayed by the kinetics of Cdc14 re-sequestration.

Next, we examined kinase-specific phospho-mutants in Cfi1/Net1. There are three known kinases that phosphorylate Cfi1/Net1 to regulate binding to Cdc14: mitotic CDKs (mainly Clb2-Cdk1), Cdc5, and Dbf2-Mob1. CDK phosphorylation of Cfi1/Net1 during early anaphase underlies FEAR network-dependent release of Cdc14 from its inhibitor (*Azzam et al., 2004*). In FEAR network mutants such as *slk19Δ*, Cdc14 release from the nucleolus and anaphase progression are delayed and are accompanied by increased cell-to-cell variability (*Figure 6A*). As reported previously (*Azzam et al., 2004*), cells harboring a *CFI1/NET1* allele with six CDK sites mutated to alanine, *cfi1/net1-6Cdk*, caused Cdc14 release defects similar to those of FEAR network mutants (*Figure 6B*).

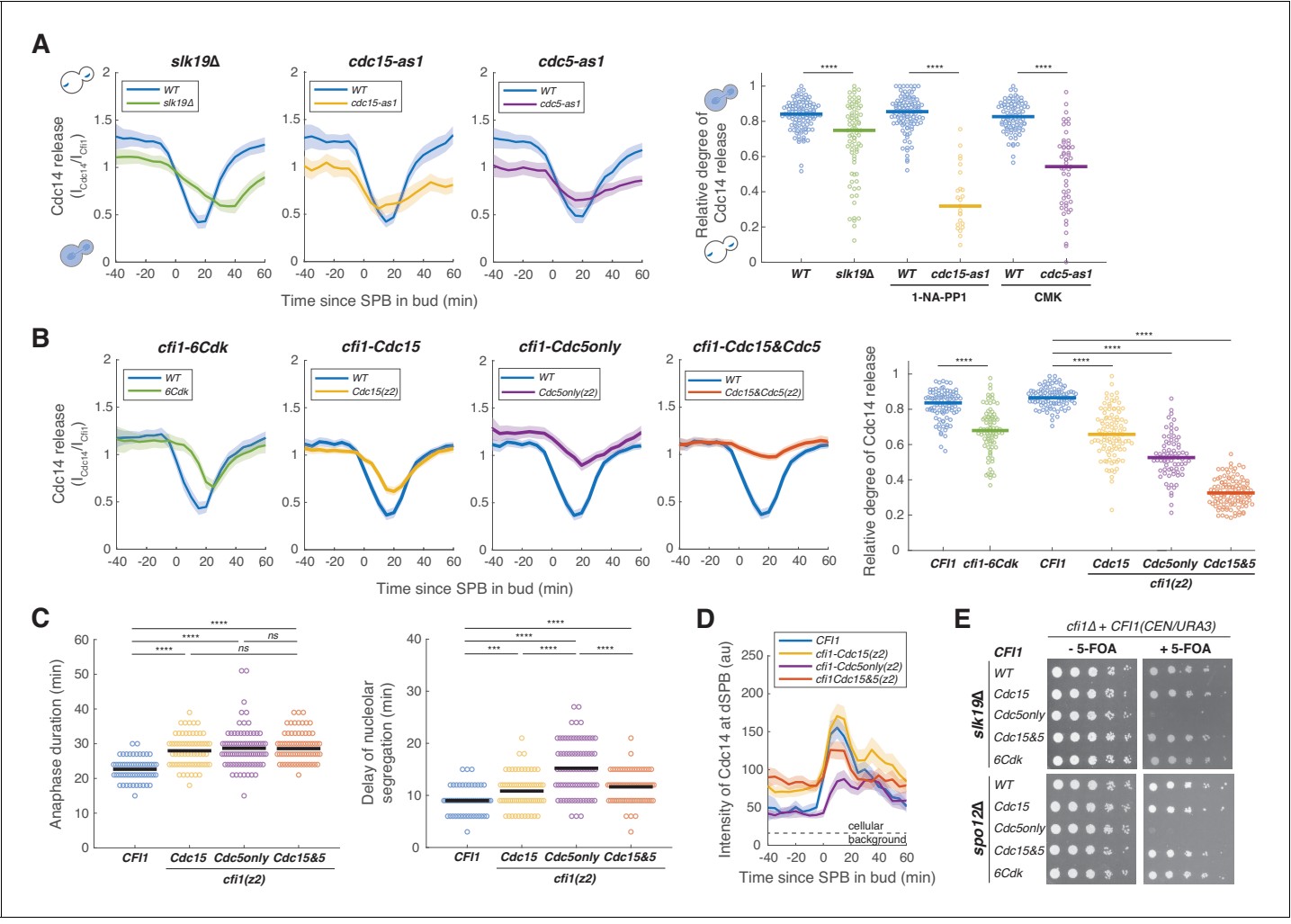

**Figure 6.** Mitotic exit network (MEN) and Cdc5 promote release of Cdc14 from the nucleolus by phosphorylating Cfi1/Net1. (**A**) Cdc14 nucleolar release kinetics in wild-type (A41387, $n$ = 134, 123, and 96 cells for each condition), *slk19Δ* (A41410, $n$ = 86 cells), *cdc15-as1* (A41408, $n$ = 38 cells), or *cdc5-as1* mutant (A41409, $n$ = 61 cells). Cells were grown at 25°C in SC medium + 2% glucose with corresponding inhibitors and imaged every 5 min for 5 hr. Release of Cdc14 from the nucleolus was quantified as the ratio of fluorescence intensity of Cdc14-eGFP to Cfi1/Net1-mScarlet-I in the nucleolus ($I_{Cdc14}/I_{Cfi1}$). Relative degree of Cdc14 release from the nucleolus was calculated with the normalized minimal Cdc14 level in the nucleolus as 1 - ($I_{Cdc14}(t_{min})/I_{Cfi1}(t_{min})$)/ ($I_{Cdc14}(t_{-20})/I_{Cfi1}(t_{-20})$), where $t_{min}$ represents the frame with minimal Cdc14 level in the nucleolus and $t_{-20}$ represents 20 min before movement of the spindle pole body (SPB) into bud. (**B**) Cdc14 nucleolar release kinetics in cells harboring wild-type *CFI1/NET1* (A41387, $n$ = 102 and 114 cells) or *CFI1/NET1* phospho-mutants for CDK sites (A41420, $n$ = 95 cells), Cdc15 sites (A41587, $n$ = 104 cells), Cdc5 sites (A41588, $n$ = 86 cells), and Cdc15&Cdc5 sites (A41589, $n$ = 131 cells). Cells were grown at 25°C in SC medium + 2% glucose and imaged every 5 min for 5 hr. (**C**) Distribution of anaphase duration and relative delay of nucleolar segregation for different *CFI1/NET1* phospho-mutants (A41436, A41590, A41591, and A41592; $n$ = 76, 85, 99, and 92 cells respectively) measured using the SPB marker Spc42-eGFP and the nucleolar marker Cfi1/Net1-mScarlet-I (see *Figure 6—figure supplement 3* for details). Cells were grown at 25°C in SC medium + 2% glucose and imaged every 3 min for 4 hr. (**D**) Intensities of Cdc14-eGFP at dSPBs in different *CFI1/NET1* phospho-mutant cells (A41387, A41587, A41588, and A41589; $n$ = 80, 82, 77, and 89 cells respectively). Cells were grown and imaged as in (**B**). (**E**) Genetic interactions between different *CFI1/NET1* phospho-mutants and *slk19Δ* (A41645, A41646, A41647, A41648, A41649) or *spo12Δ* (A41650, A41651, A41652, A41653, A41654) analyzed by plasmid shuffling (see Materials and methods for details). Fivefold serial dilutions were spotted onto plates with or without 5'-fluoroorotic acid (5-FOA) and incubated at 25°C for 2–3 days. The presence of 5-FOA selects cells that are viable after losing the *CFI1(URA3/CEN)* plasmid. For all graphs, single cell traces were aligned to the frame where the dSPB entered the bud and averaged. Solid lines represent the average. Shaded areas represent 95% confidence intervals. For distributions, each dot represents a single cell. Solid lines represent the median for (**A and B**) and the mean for (**C**). ****p<0.0001; ***p<0.001; **p<0.01; *p<0.05 by two-sided Wilcoxon rank sum test.

The online version of this article includes the following figure supplement(s) for figure 6:

**Figure supplement 1.** Phosphorylation in zone 2 of Cfi1/Net1 regulates Cdc14 release from the nucleolus.

**Figure supplement 2.** The mitotic exit network (MEN) and *CDC5* promote release of Cdc14 from the nucleolus by phosphorylating Cfi1/Net1.

**Figure supplement 3.** Cfi1/Net1 phospho-mutants delay mitotic exit and nucleolar segregation.

**Figure supplement 4.** Phosphorylation of the Cdc14 nuclear localization signal (NLS) does not play a major role in promoting mitotic exit.

*Figure 6 continued on next page*

Figure 6 continued

**Figure supplement 5.** Mutating both CDK and Cdc5 sites in Cfi1/Net1 results in severe delays in mitotic exit and nucleolar segregation.

Inactivation of the MEN using the *cdc15-as1* allele led to the previously described pattern of Cdc14 localization, where Cdc14 is initially released from the nucleolus by the FEAR network during early anaphase but is then re-sequestered in the nucleolus during later stages of anaphase (*Stegmeier et al., 2002*; *Figure 6A*). Mutating the *CDC15*-dependent phosphorylation sites in zone 2 (*cfi1-Cdc15(z2)*) resulted in a significant reduction of Cdc14 release from the nucleolus but only recapitulated ~50% of the effect of inactivating *CDC15* (compare *Figure 6A and B*). As previously reported (*Visintin et al., 2008*), inhibition of the analog sensitive *cdc5-as1* allele caused defects in both FEAR network and MEN-mediated release of Cdc14 from the nucleolus (*Figure 6A*). Mutating the *CDC5*-only phosphorylation sites in zone 2 (*cfi1-Cdc5only(z2)*) resulted in a similar reduction of Cdc14 release from the nucleolus (*Figure 6B*) as inhibiting Cdc5. Finally, combining *cfi1-Cdc5only (z2)* with *cfi1-Cdc15(z2)* (*cfi1-Cdc15&Cdc5(z2)*) caused an even greater defect in Cdc14 release from the nucleolus than either mutant alone (*Figure 6B*, *Figure 6—figure supplement 2*). These results confirmed our model where Cdc5, in addition to activating the MEN, directly phosphorylates Cfi1/Net1 to target Dbf2-Mob1 to Cfi1/Net1. Phosphorylation of Cfi1/Net1 by Dbf2-Mob1 then promotes the dissociation of Cdc14 from Cfi1/Net1.

## Phosphorylation of Cfi1/Net1 by Cdc5 and Dbf2-Mob1 promotes mitotic exit

Preventing the dissociation of Cdc14 from its inhibitor during anaphase ought to interfere with mitotic exit. Indeed, we observed a delay in all mutants analyzed (*Figure 6C*, *Figure 6—figure supplement 3*). Both *cfi1-Cdc15(z2)* and *cfi1-Cdc5only(z2)* mutant cells exited mitosis with an average delay of ~6 min (~25% increase, *Figure 6C*). In addition, consistent with Cdc5's role in the FEAR network, we observed a significant delay in nucleolar segregation in *cfi1-Cdc5only(z2)* mutant cells (*Figure 6C*). Surprisingly, *cfi1-Cdc15&Cdc5(z2)* double mutant cells which had the most severe defect in Cdc14 release from the nucleolus exhibited a similar delay in mitotic exit as the *cfi1-Cdc15 (z2)* and *cfi1-Cdc5only(z2)* single mutants and a less severe defect in nucleolar segregation compared to *cfi1-Cdc5only(z2)* (*Figure 6C*). This relatively short delay in mitotic exit is likely due to the fact that Cdc14 was not tightly sequestered in this mutant prior to anaphase. We observed elevated levels of Cdc14 at dSPB, indicating the presence of free Cdc14 not sequestered in the nucleolus, in both *cfi1-Cdc15(z2)* and *cfi1-Cdc15&Cdc5(z2)* cells prior to anaphase (*Figure 6D*, *Figure 6—figure supplement 2*). It appears that mutating *CDC15*-dependent sites in Cfi1/Net1 interferes with its ability to bind Cdc14. Nonetheless, the majority (>60%) of Cdc14 was not released during anaphase in cells harboring *cfi1-Cdc15&Cdc5(z2)*.

One possible explanation for the apparent discrepancy between the severity of the Cdc14 release defect and the more subtle delay in mitotic exit is that, in addition to targeting Cfi1/Net1, MEN also promotes cytoplasmic retention of Cdc14 by phosphorylating its NLS (*Mohl et al., 2009*). To determine the contribution of this MEN function to promoting mitotic exit in *Cfi1/Net1* mutants, we constructed a *CDC14* mutant (*cdc14-3A*) in which all three potential Dbf2-Mob1 phosphorylation sites capable of driving translocation of the NLS$_{Cdc14}$ reporter into the cytoplasm were mutated to alanine (*Figure 6—figure supplement 4A*). Interestingly, cells expressing *cdc14-3A* instead of *CDC14* exited from mitosis without significant delays (*Figure 6—figure supplement 4B*) indicating that cytoplasmic retention of Cdc14 does not play a major role in promoting mitotic exit.

Another possibility is that some MEN-dependent phosphorylation sites in Cfi1/Net1 eluded our MS analysis. However, given that Cdc14 is a potent phosphatase, we favor the idea that incomplete sequestration of Cdc14 prior to anaphase coupled with FEAR network-mediated nucleolar release of Cdc14 during early anaphase is sufficient to promote exit from mitosis with only a modest delay in the *cfi1-Cdc15&Cdc5(z2)* mutant. Consistent with this idea, we found that the *cfi1-Cdc5only(z2)* mutant, which does not display defects in sequestering Cdc14 prior to anaphase, was synthetic lethal with the FEAR network mutants *slk19Δ* and *spo12Δ* (*Figure 6E*). Furthermore, combining *cfi1-Cdc5only(z2)* with the FEAR mutant *cfi1-6Cdk* resulted in severe defects in both FEAR network and MEN mediated Cdc14 release (*Figure 6—figure supplement 5A*) and ~20 min delays in both

mitotic exit and nucleolar segregation (*Figure 6—figure supplement 5B*), the strongest defect in all mutants we have characterized. We conclude that *CDC5* and MEN-dependent phosphorylation of Cfi1/Net1 controls the protein's binding to Cdc14 and exit from mitosis.

## Discussion

As a model system for mitotic exit control and cellular signaling in eukaryotes, the MEN has been studied extensively for decades. Yet, how the MEN activates its effector Cdc14 to promote exit from mitosis has remained an enigma. Furthermore, the spatial aspect of signal transmission in the MEN, namely how a signal generated at the outer plaque of the SPBs in the cytosol reaches its target in the nucleolus, was largely unexplored. Our study provides mechanistic insights into these questions and together with prior observations lead to a model for Cdc14 regulation and mitotic exit in budding yeast (*Figure 7*).

### The role of the Polo-like kinase Cdc5 in regulating Cdc14 activation

The Polo-like kinase Cdc5 is essential for Cdc14 activation and is part of both the FEAR network and the MEN. However, the exact role(s) of Cdc5 in regulating Cdc14's nucleolar release has remained elusive due to its multiple functions in the MEN and the FEAR network. Using an allele that bypasses *CDC5*'s role in MEN activation, we revealed a novel function of Cdc5 as a priming kinase that targets Dbf2-Mob1 to its substrate Cfi1/Net1.

It was hypothesized that phosphorylation of Cfi1/Net1 by Cdc5 weakens the interaction between Cfi1/Net1 and Cdc14. Additional phosphorylation by mitotic CDKs or Dbf2-Mob1 was thought to further reduce the interaction resulting in the dissociation of Cdc14 from its inhibitor. Our results suggest an alternative model. Instead of, or at least in addition to, weakening the Cfi1/Net1-Cdc14

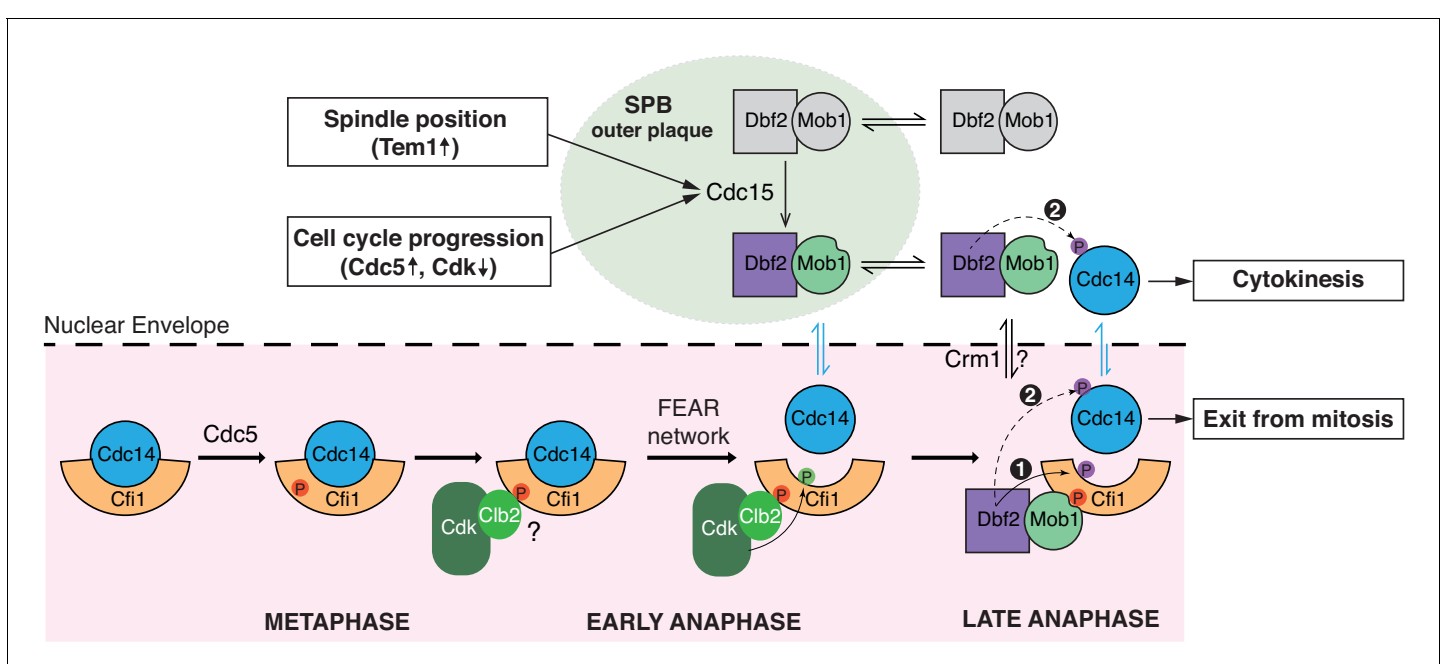

**Figure 7.** A model for Cdc14 activation and mitotic exit in budding yeast. In metaphase, Cdc5 phosphorylates Cfi1/Net1 in the nucleolus to prepare for Cdc14 release/activation in anaphase. Upon anaphase onset, the Cdc fourteen early anaphase release (FEAR) network promotes phosphorylation of Cfi1/Net1 by Clb2-Cdk1 which results in transient release of Cdc14 from the nucleolus. In the meantime, the mitotic exit etwork (MEN) kinase Cdc15 is activated by integrating inputs from both spindle position (via Tem1) and cell cycle progression (via Cdc5 and CDK activities). Activated (spindle pole body [SPB]-localized) Cdc15 phosphorylates the SPB outer plaque protein Nud1 which creates a dynamic docking site for the MEN terminal kinase complex Dbf2-Mob1 and facilitates phosphorylation and activation of Dbf2-Mob1 by Cdc15. Activated Dbf2-Mob1 gains access to the nucleus and is targeted to the nucleolus by interacting with Cdc5-primed Cfi1/Net1. Nucleolar Dbf2-Mob1 then phosphorylates Cfi1/Net1, keeping Cdc14 dissociated from its nucleolar inhibitor to trigger exit from mitosis. In addition, active Dbf2-Mob1 in the nucleolus and/or cytoplasm phosphorylates Cdc14 at its nuclear localization signal (NLS) resulting in cytoplasmic retention of Cdc14 to facilitate cytokinesis.

interaction, Cdc5 phosphorylation targets Dbf2-Mob1 to Cfi1/Net1. Mutating the Cdc5-only phosphorylation sites in Cfi1/Net1 abolished nucleolar enrichment of Dbf2-Mob1 and resulted in a reduction in MEN-mediated dissociation of Cdc14 from Cfi1/Net1. Interestingly, this mutation also caused severe defects in the FEAR network mediated release of Cdc14 from Cfi1/Net1. We thus propose that Cdc5 priming phosphorylation is required not only for Dbf2-Mob1 to phosphorylate Cfi1/Net1 but also for mitotic CDKs (*Figure 7*). Consistent with this hypothesis, we found that most Cdc5 phosphorylation sites in Cfi1/Net1 are already phosphorylated by Cdc5 in metaphase (*Figure 5—figure supplement 3E*). This observation indicates that the docking site(s) on Cfi1/Net1 for Dbf2-Mob1 (and mitotic CDKs) is already present in metaphase prior to the activation of the FEAR network and the MEN (*Figure 7*). This model is further supported by the observation that inhibition of Cdc5 eliminates CDK mediated phosphorylation of T212 in Cfi1/Net1 (*Azzam et al., 2004*). Cdc5 activity is regulated by the DNA damage checkpoint (*Cheng et al., 1998*; *Sanchez et al., 1999*). We speculate that making FEAR network and MEN-mediated release of Cdc14 from the nucleolus dependent on Cdc5's priming activity ensures that DNA damage has been repaired and the checkpoint silenced prior to exit from mitosis.

## The FEAR network and the MEN regulate Cdc14 binding to Cfi1/Net1 by different mechanisms

Although both mitotic CDKs and Dbf2-Mob1 appear to require *CDC5*-dependent priming phosphorylation of Cfi1/Net1, the mechanism whereby mitotic CDKs and Dbf2-Mob1 disrupt the interaction between Cfi1/Net1 and Cdc14 is quite different. Mitotic CDKs phosphorylate Cfi1/Net1 mainly on S166, T212, S252, T297, and T304; Dbf2-Mob1 targets sites S259, S295, S362, S439, and S497 (*Figure 5A*, *Supplementary file 4*). Given that mitotic CDKs and Dbf2-Mob1 target different sites, we propose that increasing the acidity of aa160–500 within Cfi1/Net1 rather than site-specific phosphorylation disrupts the interaction between Cdc14 and its inhibitor. Cfi1/Net1 is an integral part of the nucleolus, which has recently been described as a phase-separated structure (*Feric et al., 2016*; *Shin and Brangwynne, 2017*). Overall phosphorylation rather than phosphorylation of specific sites has been shown to disrupt interactions within such structures (*Carpenter et al., 2018*; *Owen and Shewmaker, 2019*). Perhaps extraction of Cdc14 from the nucleolar phase requires a similar mechanism.

## The MEN as a model for cross-compartment signaling

The MEN, most closely related to the Hippo pathway, employs most, if not all, of the principles governing classic receptor tyrosine signaling logic to convey a signal generated at SPBs to the MEN effector Cdc14 in the nucleolus: (1) scaffold-assisted signaling (at the SPB), (2) signal transmission across organelle boundaries – from the cytoplasm to the nucleus, and (3) activation of the effector in a sub-compartment (the nucleolus). As such, we believe that the molecular mechanisms governing MEN activity are broadly applicable to intracellular signal transmission in general.

## (1) Dynamic scaffold-assisted signaling

Scaffold-assisted assembly of signaling complexes is a widespread phenomenon in eukaryotic signal transduction cascades (*Good et al., 2011*). We find that in the MEN, assembly of Cdc15-(Dbf2-Mob1) signaling complex on the scaffold Nud1 is highly dynamic and this dynamicity is crucial for effector activation. We propose that this dynamicity also serves to amplify the signal. Cdc15 is the limiting enzyme of the pathway: it is the least abundant component of the MEN and hyperactivating Cdc15 increases Dbf2-Mob1's kinase activity by >40 fold (*Rock and Amon, 2011*). We further hypothesize that the relatively low affinity of Mob1 for phosphorylated-Nud1 ($K_d$ = 2.4 µM; note that for an optimal Mob1 binding phosphopeptide the $K_d$ is 174 nM) (*Rock et al., 2013*) is selected for to facilitate the fast turnover rate of Dbf2-Mob1 at SPBs and thus to promote release of the kinase and signal amplification. In this model, the binding affinity/kinetics of kinases to their signaling scaffolds is an important parameter that cells fine-tune to generate desirable signaling properties of scaffold-assisted signaling pathways.

## (2) Regulated compartment access

Many signals, be they generated outside or within the cell, ultimately, result in a nuclear response. As such, signals have to be propagated from the cytoplasm into the nucleus. Our studies have led to the discovery that in the MEN, this nuclear access is cell cycle regulated. Prior to anaphase, Dbf2-Mob1 is actively exported out of the nucleus by Crm1 through the conserved NES within the N-terminus of Dbf2. Upon MEN activation, nuclear partitioning of Dbf2-Mob1 increases, likely a result of both increased nuclear import through modification of Mob1's inhibitory N-terminus and decreased nuclear export through modification of Dbf2. One potential mechanism for the latter is via phosphorylation of Dbf2's NES at S17 and S20, possibly by Cdc15. Our observation that mutating these two sites exacerbated the temperature sensitivity of *cdc15-2* (*Figure 3—figure supplement 3C*) is consistent with this possibility.

Disrupting the NES resulted in an increase in nuclear/nucleolar localization of Dbf2-Mob1 in all cell cycle stages, interestingly, including anaphase (*Figure 3E*). This increase of nuclear localization in anaphase suggests that only a small fraction of Dbf2-Mob1 is activated by Cdc15 to enter the nucleus at any given time during anaphase. Active Dbf2-Mob1 is also needed in the cytosol to phosphorylate substrates other than Cfi1/Net1 such as those involved in cytokinesis. We speculate that fine-tuning the balance of nuclear versus cytosolic Dbf2-Mob1, possibly through maintaining a dynamic shuttling of active Dbf-Mob1 between the nucleus and cytoplasm, is important for the timing of late cell cycle events. The dynamic shuttling of Dbf2-Mob1 in combination with the relatively small fraction of active Dbf2-Mob1 would also explain the absence of visible nuclear translocation of Dbf2-Mob1 upon activation. Interestingly, mammalian Dbf2, known as LATS and is thought to mainly function in the cytosol (*Yu and Guan, 2013*), has been found to localize to the nucleus (*Britschgi et al., 2017*; *Li et al., 2014*), indicating nuclear shuttling of kinases might play a role in Hippo signaling as well.

## (3) Substrate targeting by priming phosphorylation

Upon entry into the nucleus, Dbf2-Mob1 specifically functions in the nucleolus to promote the dissociation of Cdc14 from its inhibitor Cfi1/Net1. Priming phosphorylation by Cdc5 on Cfi1/Net1 ensures that Dbf2-Mob1 executes this function effectively. Dbf2-Mob1 binds to Cdc15-phosphorylated Nud1 through Mob1's phosphoserine-threonine binding domain (*Rock et al., 2013*). We demonstrated that a similar mechanism mediates the interaction between Dbf2-Mob1 and Cdc5-phosphorylated Cfi1/Net1 (*Figure 5G*). Interestingly, during cytokinesis, phosphorylation of Dbf2-Mob1's substrate Hof1 by Cdc5 has also been shown to facilitate the binding of Hof1 to Mob1 (*Meitinger et al., 2011*; *Rock et al., 2013*). Based on our observation that a quarter of potential Dbf2-Mob1 substrates are also targets of Cdc5 (*Figure 5—figure supplement 1E*) we further speculate that priming phosphorylation by Cdc5 is a general mechanism for targeting Dbf2-Mob1 to its substrates.

It is worth noting that Cdc5's consensus motif with an acidic residue at the −2 position of the pS/T (*Kettenbach et al., 2011*; *Nakajima et al., 2003*) does not fit well with the optimal Mob1 binding motif which prefers Y/F at the −2 position of the pS/T (*Rock et al., 2013*). This is likely beneficial as a low affinity between Dbf2-Mob1 and its substrates would increase substrate turnover and prevent sequestration of active Dbf2-Mob1. This low affinity would also explain the weak and transient localization of Dbf2-Mob1 in the nucleolus that evaded detection previously. Mob1's phosphoserine-threonine binding domain is well conserved from yeast to humans (*Rock et al., 2013*) suggesting that substrate targeting through priming phosphorylation might occur for LATS-MOB1 in the Hippo pathway as well.

Priming phosphorylation is a widely used mechanism to ensure effective kinase action at a particular site in the cell. Perhaps the best studied example for priming is the Polo-like kinases whose polo-box domain directs the kinase to specific subcellular structures and substrates that have been previously phosphorylated by a priming kinase such as CDKs (*Elia et al., 2003*; *Lowery et al., 2005*). Although MAP kinases are mainly directed to their substrates through specialized docking motifs (*Bardwell, 2006*; *Cargnello and Roux, 2011*) without priming, it has been demonstrated that successive phosphorylation through priming could contribute to the sensing of MAPK signal duration and strength (*Murphy et al., 2002*). These examples together with our findings for Dbf2-Mob1 underscore the importance of priming phosphorylation as a conserved paradigm for achieving specificity and efficiency in cellular signal transduction.

# Materials and methods

**Key resources table**

| Reagent type (species) or resource | Designation | Source or reference | Identifiers | Additional information |
|---|---|---|---|---|
| Gene (*S. cerevisiae*) | See *Supplementary file 1* | | | |
| Strain, strain background (*S. cerevisiae*) | W303 | https://www.yeastgenome.org/strain/w303 | | |
| Genetic reagent (*S. cerevisiae*) | See *Supplementary file 1* | | | |
| Recombinant DNA reagent | See *Supplementary file 2* | | | |
| Strain, strain background (*E. coli*) | DH5α | New England Biolabs | Cat# C2987U | Chemical competent cells |
| Antibody | Anti-GFP [JL-8] (Mouse monoclonal) | Takara Bio | Cat# 632381; RRID:AB_2313808 | WB (1:1000) |
| Antibody | Anti-Myc [9E10] (Mouse monoclonal) | Abcam | Cat# ab32; RRID:AB_303599 | WB (1:500) |
| Antibody | Anti-V5 (Mouse monoclonal) | Invitrogen | Cat# R960-25; RRID:AB_255656 | WB (1:2000) |
| Antibody | Anti-Kar2 (Rabbit polyclonal) | Gift from Mark Rose | N/A | WB (1:200,000) |
| Antibody | HRP-conjugated anti-mouse IgG (Sheep monoclonal) | GE | Cat# NA9310; RRID:AB_772193 | (1:10,000) |
| Antibody | HRP-conjugated anti-rabbit IgG (Donkey monoclonal) | GE | Cat# NA934; RRID:AB_772206 | (1:10,000) |
| Antibody | Anti-tubulin [YOL1/34] (Rat monoclonal) | Abcam | Cat# Ab6161; RRID:AB_305329 | IF (1:100) |
| Antibody | FITC-anti-Rat IgG (Donkey polyclonal) | Jackson ImmunoResearch | Cat# 712-095-153; RRID:AB_2340652 | (1:50) |
| Peptide, recombinant protein | Streptavidin | Sigma-Aldrich | Cat# 189730 | |
| Peptide, recombinant protein | α-factor | The Koch Institute Swanson Biotechnology Center – Biopolymers Core Facility | N/A | |
| Commercial assay or kit | Gibson Assembly Master Mix | New England Biolabs | Cat# E2611S | |
| Commercial assay or kit | Q5 Site-Directed Mutagenesis Kit | New England Biolabs | Cat# E05545 | |
| Commercial assay or kit | Bradford protein assay | BioRad | Cat# 5000006 | |
| Commercial assay or kit | BCA protein assay | Pierce | Cat# 23227 | |
| Commercial assay or kit | MyOne Streptavidin C1 dynabeads | Thermo Fisher Scientific | Cat# 65001 | |
| Chemical compound, drug | Nocodazole | Sigma-Aldrich | Cat# M1404 | |
| Chemical compound, drug | Biotin | Sigma-Aldrich | Cat# B4639 | |
| Chemical compound, drug | 1-NA-PP1 | Cayman Chemical | Cat# 10954 | |
| Chemical compound, drug | CMK | MedChem Express | Cat# HY-52101 | |

*Continued on next page*

*Continued*

| Reagent type (species) or resource | Designation | Source or reference | Identifiers | Additional information |
|---|---|---|---|---|
| Chemical compound, drug | Phycocyanobilin (PCB) | Santa Cruz Biotechnology | Cat# sc-396921 | |
| Software, algorithm | FIJI (ImageJ) | *Schindelin et al., 2012* | RRID:SCR_002285 | |
| Software, algorithm | MATLAB_R2018b | Mathworks, Inc (2018) | RRID:SCR_001622 | |
| Software, algorithm | SnapGene v4.3 | https://www.snapgene.com | RRID:SCR_015052 | |
| Software, algorithm | Custom MATLAB codes | This paper | https://github.com/snow-zhou/Dbf2-Mob1 (copy archived at swh:1:rev:edb372c2e4ddf8eb2278536a7fa580abaa60acf1) | |
| Other | Mini Bead Beater | Biospec Products | N/A | |

## Construction of yeast strains and plasmids

All *Saccharomyces cerevisiae* yeast strains used in this study are derivatives of W303 (A2587) and are listed in *Supplementary file 1*. All plasmids used in this study are listed in *Supplementary file 2*. Yeast were cultured in standard YEP media (1% yeast extract, 2% peptone) with 2% D-glucose, or in standard Synthetic Complete (SC) media with either 2% D-glucose or 2% raffinose prior to induction of *GAL1/10* promoters with 1% raffinose + 1% galactose as indicated in the figure legends. Cells were cultured at 25°C unless noted otherwise.

C-terminal fusions were constructed using standard PCR-based methods. N-terminal truncation of *MOB1* mutants were made by first constructing the truncations with N-terminal eGFP fusion in plasmids (pA2828-2830) based on pRS305H-*MOB1* (*König et al., 2010*). PCR products of the fusion with 500 bp upstream promoter region and 300 bp downstream of the *MOB1* ORF as well as the hphMX6 marker were transformed and integrated at the *MOB1* locus. Correct integration and mutations were confirmed by PCR and sequencing. The NLS$_{Cdc14}$ reporter was made by Gibson assembly. NLS$_{Cdc14}$ (Cdc14aa450-551) from pA2725 (*Mohl et al., 2009*) and synthesized yeast codon optimized miRFP670 (*Shcherbakova and Verkhusha, 2013*) (denoted as ymiRFP670) were assembled into the p404TEF1 vector (pA2726) which yielded pA2786 for integration at the *trp1* locus. The PhyB constructs were integrated at the *leu2* locus as single integrants (*Yang et al., 2013*). The eGFP-PIF C-terminal tagging plasmid (pA2821) was made by replacing the mCitrine in pA2721 with eGFP and correcting the frame shift mutation (missing a G at codon#2) in the *natMx6* coding sequence in the original plasmid acquired from Addgene. The TurboID tagging plasmid (pA2859) was constructed by inserting the TurboID-V5 sequence from pA2847 (*Branon et al., 2018*) into the vector backbone of pA2821 between the linker and natMx6.

Point mutations and truncations in the *GAL-DBF2* plasmids were made using Q5 site-directed mutagenesis (NEB). Single point mutation in *DBF2* (*dbf2-L12A*) was introduced at the endogenous locus using Cas9-mediated gene editing based on previously published protocols (*Anand et al., 2017*) with modifications. Briefly, BplI cut plasmid pA2911 containing Cas9 and gRNA missing the 20 bp target specific complementary sequence was transformed into A2587 together with (1) a ~250 bp synthesized double-stranded DNA (IDT) containing the 20 bp complementary sequence for *DBF2* (TGCTCATATTGCCTGCCAAG) sandwiched with homologous sequences to the gRNA sequence in plasmid pA2911 for gap repair, and (2) an 80mer single-stranded DNA harboring *dbf2-L12A* mutation as the repair template. Successfully edited clones were checked by sequencing and cured of the Cas9 plasmid. Phospho-mutants of *DBF2* (*dbf2-S17,20A* and *dbf2-S17,20D*) were first constructed in plasmids (pA2948, pA2953, and pA2954) by Q5 site-directed mutagenesis. The *DBF2* ORF as well as the *URA3* marker were then PCR amplified and transformed into A32629 to integrate at the *DBF2* locus as the sole copy of *DBF2*. Correct integration and mutations were confirmed by PCR and sequencing.

To generate *CFI1/NET1* phospho-mutants, a fusion of wild-type *CFI1/NET1* and mScarlet-I with ~300 bp upstream and downstream regions of *CFI1/NET1* were first Gibson assembled into a single-integration vector backbone to generate plasmid pA2898. *CFI1/NET1* phospho-mutants were synthesized (Twist Bioscience) in two segments (aa25-624 and aa617-1173) and Gibson assembled

into the vector backbone of pA2898 to replace wild-type *CFI1/NET1*. These *CFI1/NET1* constructs were integrated into strain A41362 harboring a heterozygous deletion of *CFI1/NET1* and haploid strains with *cfi1Δ* and different phospho-mutant forms of *CFI1/NET1* integrated as a single copy at the *leu2* locus as the sole source of *CFI1/NET1* were obtained through tetrad dissection. We attempted to generate phospho-mimetic mutants of *CFI1/NET1* where either the Cdc15-dependent or Cdc5 only sites in zone 2 were replaced with the Asp or Glu to mimic phosphorylation (A41695-A41706). However, as observed with many other proteins such as Cdc15 phosphorylation of Nud1 (*Rock et al., 2013*), these mutations did not effectively mimic phosphorylation and in the case of Cdc5only sites led to a loss of function with respect to Mob1 localization to the nucleolus.

Phospho-mutants in the NLS$_{Cdc14}$ reporter were made by first introducing the point mutations into the integration plasmid (pA2735) with Q5 site-directed mutagenesis and then transformed and integrated at the *ura3* locus. The triple mutants of *CDC14* (S531,537,546A) were introduced at the endogenous locus in A2588 using Cas9-mediated gene editing as described above. The 20 bp complementary sequence for the gRNA used to target *CDC14* was GGAGAGTAACGTCAGGGAGA. Ectopically expressed *mob1Δ132-eGFP* and *mob1Δ132-3RA-eGFP* were made by removing the first 132aa of Mob1 from pA2124 and pA2126 (*Rock et al., 2013*) with Q5 site-directed mutagenesis, generating pA2972 and pA2973. These plasmids were integrated at the *trp1* locus and were expressed under the endogenous *MOB1* promoter in the presence of a wild-type copy of *MOB1*.

### FRAP analysis

FRAP analysis was performed on a DeltaVision-OMX Super-Resolution Microscope (Applied Precision, GE Healthcare Bio-Sciences) using a 60× oil objective and a 488 nm laser adjusted to bleach an area of approximately 0.5 μm in radius. Two prebleach images were acquired followed by a laser pulse (100% intensity) of 0.02 s duration and postbleach images were acquired at 1 s per frame for 30 s. Images at each time points were maximum projections of 7 *z* stacks with 0.5 μm spacing. Images were analyzed with a custom MATLAB script. After subtracting the background, fluorescence intensities in the cytosol, $I_{cytosol}$ (*t*), and at the SPB, $I_{SPB}$ (*t*) were measured after segmenting the cell and SPBs. Photobleaching was corrected by normalizing $I_{SPB}$ (*t*) with $I_{cytosol}$ (*t*), $I_{SPB\_norm}$ (*t*) = $I_{SPB}$ (*t*) / $I_{cytosol}$ (*t*). Double normalization for FRAP was calculated to scale the photobleaching effect between 0 and 1: $I_{SPB\_FRAP}$ (*t*) = [$I_{SPB\_norm}$ (*t*) − $I_{SPB\_norm}$ (0)] / [$I_{SPB\_norm}$ (*pre*) − $I_{SPB\_norm}$ (0)], where *t* = 0 is the time point (frame) right after photobleaching and *t* = *pre* is the time point (frame) right before photobleaching. The double normalized FRAP curves were then fitted to a single exponential curve: $y = y_{max}\left(1 - e^{\left(\frac{-ln2}{t_{1/2}}\right)t}\right)$, where $y_{max}$ is the fraction recovered while $t_{1/2}$ is the half-recovery time.

### Microscopy and image analysis

For live-cell microscopy, cells were imaged on agarose pads (2% agarose in SC medium + 2% glucose, unless otherwise noted) affixed to a glass slide and covered with a coverslip. Imaging was performed on a DeltaVision Elite platform (GE Healthcare Bio-Sciences) with an InsightSSI solid state light source, an UltimateFocus hardware autofocus system and a model IX-71, Olympus microscope controlled by SoftWoRx software. A 60× Plan APO 1.42NA objective and CoolSNAP HQ2 camera were used for image acquisition. For each time point, 7 *z* sections with 1 μm spacing were collected for each channel and were deconvolved. Maximum projections of the deconvolved *z* stack were used for fluorescence quantification.

Image analysis was performed with custom scripts in MATLAB. First, yeast cells were segmented and tracked through time using the bright-field image stacks and a previously reported algorithm (*Ricicova et al., 2013*). A few modifications were made for the tracking process. Images were first aligned to correct for drift in the *xy* plane and cell segmentations in the last frame of the time-lapse series were used as the reference for tracking. Next, fluorescence images of cell cycle markers (such as Spc42 and or Cfi1/Net1) were segmented and tracked based on cell segmentation. Appearance of a cell cycle marker in a cell during the acquisition period was used to identify buds (daughter cells) and cell division events. Tracking of the cell cycle markers that migrated into the buds were used to identify the corresponding mother cells. Finally, for each division event identified, localization of Mob1 (or Cdc14) at regions defined by the cell cycle markers was quantified.

For Mob1 localization at the SPBs ($I_{SPB}$), maximum intensity of Mob1 at SPBs (dilated from the SPB area based on segmentation of Spc42) was used given that the size of SPBs (~100 nm) is within the diffraction limit of light microscopy. For Mob1 localization in the nucleolus ($I_{nucleolus}$), the median intensity of Mob1 in the nucleolus as segmented by Cfi1/Net1 was used. To calculate the relative enrichment of Mob1 at the SPB or in the nucleolus, Mob1 intensities at these sites as defined above were normalized to the median intensity of Mob1 in the cytosol which was defined as the cell area with cell cycle marker area subtracted ($I_{cytosol}$) : $I_{SPB\ or\ nucleolus}\ /\ I_{cytosol} - 1$. For the Dbf2-Mob1 kinase activity reporter, $NLS_{Cdc14}$, its translocation from the nucleus into the cytoplasm was quantified with the coefficient of variation (CV) of the pixel intensities within the dividing cell. CV is defined as the standard deviation divided by the mean.

To quantify Cdc14 release from the nucleolus, the ratio of Cdc14 intensity to Cfi1/Net1 intensity $\left(I_{Cdc14}/I_{Cfi1}\right)$ was calculated for each pixel within the nucleolus as segmented using Cfi1/Net1 and averaged.

Single cell traces were aligned based on the timing of anaphase onset or the movement of a SPB into the bud as indicated in figures and averaged. 95% confidence intervals were calculated as $\mu \pm 1.96 * \sigma/\sqrt{n}$, where $\mu$ and $\sigma$ denote the mean and standard deviation respectively and $n$ is the number of cells measured.

## Immunoblot analysis

Log-phase cultures of cells grown in YEP + 2% glucose were harvested and treated with 5% tri-chloroacetic acid (TCA) at 4℃ overnight. TCA treated cell pellets were washed with acetone, air dried, and resuspended in lysis buffer (10 mM Tris, 1 mM EDTA, 2.75 mM DTT, pH = 8). Cells were lysed by bead-beating using a Multivortexer (max speed, 20 min) and glass beads at 4℃ and followed by boiling in SDS PAGE protein loading buffer for 5 min. Lysates were clarified by centrifugation and were resolved on a 15-well NuPAGE 4–12% Bis-Tris protein gel (Thermo Fisher Scientific) prior to transfer onto nitrocellulose membranes. GFP-Mob1 and variants were detected using an anti-GFP antibody (Clontech, JL-8) at a 1:1000 dilution. Nud1-13myc and Cfi1/Net1-3myc were detected using an anti-Myc antibody (Abcam, 9E10) at a 1:500 dilution. Mob1-V5-TurboID was detected using an anti-V5 antibody (Invitrogen) at a 1:2000 dilution. Kar2 was detected using a rabbit anti-Kar2 antiserum at a 1:200,000 dilution. Secondary antibodies were used at a 1: 10,000 dilution. Blots were imaged using the ECL Plus system (GE Healthcare).

## TurboID-MS and streptavidin gel-shift assay

To identify biotinylated proteins as a result of interaction or physical proximity with TurboID tagged bait protein by mass spectrometry (MS), log-phase cells with TurboID tagged bait and untagged control cells were grown in YEP + 2% glucose + 50 μM biotin at room temperature for 3.5 hr (approximately two doublings). Approximately 40 OD of cells were harvested for each sample and were treated with 5% TCA at 4℃ for a minimum of 10 min. TCA treated cells were pelleted, washed with 50 mM Tris (pH = 7.5) and acetone, and dried. Dried cell pellets were resuspended in lysis buffer (50 mM Tris, pH 7.5, 1 mM EDTA, 5 mM DTT, 1 mM PMSF, and complete mini protease inhibitor cocktail by Roche) and were lysed by bead-beating with chilled MiniBeadbeater (Biospec) and glass beads for 5 min followed by boiling in 1% SDS for 5 min. Lysates were diluted with RIPA buffer (50 mM Tris, pH 7.5, 150 mM NaCl, 0.1% SDS, 0.5% sodium deoxycholate, 1% NP40) and clarified by centrifugation. Protein concentration of the lysates was measured by Bradford assay. 350 μl of MyOne streptavidin C1 dynabeads (Thermo Fisher Scientific) were washed twice with RIPA buffer, incubated with clarified lysates containing ~3 mg of total protein for each sample with rotation for 1 hr at room temperature, then moved to 4℃ and incubated overnight with rotation. On the second day, the supernatants (flow through) were removed and the beads were washed twice with 1 ml of 0.1 M $Na_2CO_3$, once with 1 ml of 2 M urea in 10 mM Tris (pH = 7.5), and twice with 1 ml RIPA buffer. Bound proteins were eluted by boiling the beads in 30 μl of 3× protein loading buffer supplemented with 2 mM biotin. Small aliquots of samples were saved along the process to monitor for the enrichment for biotinylated proteins by immunoblotting for V5 (included in the TurboID tagged bait) and biotin.

Eluted proteins were resolved on a 10-well NuPAGE 4–12% Bis-Tris protein gel (Thermo Fisher Scientific), stained with Coomassie (Imperial Protein Stain, Thermo Fisher Scientific) and entire gel

lanes were excised and cut into 1 mm pieces. Proteins were reduced with 20 mM dithiothreitol (Sigma) for 1 hr at 56°C and then alkylated with 60 mM iodoacetamide (Sigma) for 1 hr at 25°C in the dark. Proteins were then digested with 12.5 ng/µl modified trypsin (Promega) in 50 µl 100 mM ammonium bicarbonate, pH 8.9 at 25°C overnight. Peptides were extracted by incubating the gel pieces with 5% formic acid in 50% acetonitrile and then 100 mM ammonium bicarbonate, repeated twice, followed by incubating the gel pieces with 100% acetonitrile and then 100 mM ammonium bicarbonate, repeated twice. Each fraction was collected, combined, and reduced to near dryness in a vacuum centrifuge. Peptides were desalted using C18 SpinTips (Protea, Morgantown, WV).

Peptides were separated by reverse phase HPLC (Thermo Easy nLC1000) using a precolumn (made in house, 6 cm of 10 µm C18) and a self-pack 5 µm tip analytical column (12 cm of 5 µm C18, New Objective) over a 140 min gradient before nanoelectrospray using a QExactive HF-X mass spectrometer (Thermo). Raw mass spectral data files (.raw) were searched using Proteome Discoverer (Thermo) and Mascot version 2.4.1 (Matrix Science). Mascot search parameters were: 10 ppm mass tolerance for precursor ions; 15 mmu for fragment ion mass tolerance; two missed cleavages of trypsin; fixed modification was carbamidomethylation of cysteine; variable modifications were methionine oxidation and lysine biotinylation. Only peptides with a Mascot score greater than or equal to 25 and an isolation interference less than or equal to 30 were included in the data analysis.

There are several endogenous biotinylated proteins in yeast. To identify specific proteins enriched in the TurboID labeling experiments (hits), total peptides identified for each protein in cells with TurboID tagged bait were compared with the unlabeled control cells to calculate the ratio of enrichment. A threshold of 10 standard deviations above the average ratio of enrichment for all proteins in each sample was used to identify hits (see *Figure 2—figure supplement 1B*).

For the streptavidin gel-shift assays, cell lysates were prepared as for typical immunoblotting experiments. Prior to loading samples onto protein gels, 20 µl of lysates for each sample was incubated with 2 µl of 10 mg/ml streptavidin for 10 min at room temperature with rotation. Treated samples were then resolved by SDS-PAGE gel at 4°C and processed for immunoblotting as described above.

## PhyB-PIF based optogenetics

The PhyB-PIF based optogenetics experiments were performed based on previous reports (*Jost and Weiner, 2015*; *Yang et al., 2013*). Cells grown to log phase in 1× SC medium + 2% glucose were incubated with 31.25 µM (0.5 µl of 12.5 mM stock in 200 µl culture) Phycocyanobilin (PCB, Santa Cruz Biotechnology) for 2 hr at room temperature in the dark. Cells were then pelleted and resuspended in fresh medium without PCB and were mounted onto an agarose pad with 2× SC medium + 2% glucose for imaging. To apply the light, we attached one 650 nm and one 750 nm light-emitting diode (LED, Light-speed Technologies) onto the microscope condenser. Light was controlled manually during the experiments. For the continuous exposure of 650 nm light during time-course experiments, the light was briefly turned off for each image acquisition. To quantify the relative enrichment of target protein in the PhyB-anchored region, pixel intensities of the target protein (Mob1-eGFP-PIF, $I_{PIF}$) and PhyB (PhyB-mCherry, $I_{PhyB}$) inside the cell were fitted to a line ($I_{PIF} = \alpha + \beta I_{PhyB}$). The slope of the fitted line ($\beta$) was used to assess the extent of co-localization or enrichment of the target protein in the PhyB-anchored region. This method is robust against both photobleaching of the target protein during the time-course and difference in the shape and size of the anchored region (the nucleolus) at different cell cycle stages.

## Phosphoproteomics

To map phosphorylation events that depend on *CDC5* or *CDC15* (MEN) activity, we used the analog-sensitive alleles of *CDC5* (*cdc5-as1*) or *CDC15* (*cdc15-as1*) and compared phosphopeptides with and without inhibiting the kinase in anaphase (and metaphase for Cdc5) enriched cultures. To enrich for anaphase cells, we first synchronized cells in G1 with α-factor (5 µg/ml for 2.5 hr). α-factor arrested cultures were then washed and released into YEP + 2% glucose + inhibitor (5 µM CMK for *CDC5*/*cdc5-as1* or 10 µM 1-NA-PP1 for *CDC15*/*cdc15-as1*) at room temperature to progress to anaphase (~100 min). Synchronization and cell cycle stage were assessed by monitoring budding and spindle length at various time points. Spindles were visualized by immunofluorescence using an anti-tubulin antibody (Abcam ab6161). Cultures collected for the anaphase experiment harbored 74%

(*WT* + 1-NA-PP1), 95% (*cdc15-as1*, 1-NA-PP1), 69% (*WT* + CMK), and 94% (*cdc5-as1* + CMK) cells with anaphase spindles.

Phosphopeptide preparation was based on previous methods (*Holt et al., 2009*; *Villén and Gygi, 2008*). Approximately 25 ODs of cells were harvested for each replicate (six replicates per samples for the anaphase experiment and three replicates per sample for the metaphase experiment) by centrifugation (5 min at 3000 rpm). Cell pellets were washed with cold 50 mM Tris (pH = 7.5) and resuspended in cold lysis buffer containing 8 M urea in 50 mM Tris (pH = 7.5), 75 mM NaCl, 50 mM NaF, 50 mM β-glycerophosphate, 1 mM sodium orthovanadate, 10 mM sodium pyrophosphate and protease inhibitor cocktail (complete mini, EDTA-free, Roche). Cells were lysed under denaturing conditions by bead-beating with a chilled MiniBeadbeater (Biospec, two rounds of three cycles of 90 s each) and glass beads. The protein extract was then separated from the beads and insoluble material by centrifugation. Protein concentration of the lysate was determined by BCA protein assay (Pierce).

The protein extraction was ultrasonically lysed at 4°C for 2 min with six rounds using a VialTweeter device (Hielscher-Ultrasound Technology). The proteins extracted were then processed with a precipitation-based digestion protocol using trypsin (Promega) at a ratio of 1:20 as described previously (*Collins et al., 2017*). About 500 μg of resultant peptides from each sample was subjected for phosphopeptide enrichment, using the High-Select Fe-NTA kit (Thermo Scientific, A32992) according to the kit instruction, as described previously (*Gao et al., 2019*). About 1.5 μg of phosphopeptides enriched from each sample was used for DIA-MS measurement on an Orbitrap Fusion Lumos Tribrid mass spectrometer (Thermo Scientific) platform coupled with a nanoelectrospray ion source, as described previously (*Mehnert et al., 2019*). DIA-MS data analyses were performed using Spectronaut v13 (*Bruderer et al., 2017*) using default setting, by searching against a database of 6632 yeast open reading frames (*Hebert et al., 2014*) and the Swiss-prot yeast proteome database. In particular, the PTM localization option in Spectronaut v13 was enabled to locate phosphorylation sites (*Bekker-Jensen et al., 2020*; *Collins et al., 2017*), with the probability score cutoff >0.75(7), resulting Class-I peptides to be identified and quantified. Both peptide and protein FDR cutoff (Q value) were controlled at 1%, and the label-free protein quantification of proteome and phospho-proteome was performed using the default settings in Spectronaut.

To identify peptides whose phosphorylation depended on *CDC5* or *CDC15*, we calculated the ratio as well as the p-value of peptide intensities in *WT* versus *as1* samples. The p-values were calculated using a two-sided Student's *t*-test and were adjusted for false discovery rate (FDR) of multiple hypothesis testing using the linear step-up procedure (Benjamini–Hochberg procedure). A threshold of ratio $R > 2$ and $p_{adj} < 0.05$ was used to identify hits. Additional hits were also included for peptides that were not defected in the *as1* samples but were detected in at least five out of seven replicates (or four out of six replicates in the metaphase experiment) of the *WT* samples. Peptides that were only detected in one replicate of the *as1* samples but were detected in at least five out of seven replicates (or four out of six replicates in the metaphase experiment) of the *WT* samples with a ratio >2 were also included. After we have identified hits for peptides, we mapped the peptides and phosphorylation sites in those peptides to proteins in the yeast proteome. For phosphorylation sites that were detected in multiple peptides, peptides with a single phosphorylation were given priority. We marked a site as a strong hit only when all single phosphorylation peptides for the specific site fit our selection criteria. If a site was only detected in multi-phosphorylation peptides that fit our selection criteria, we designated that site as a weak hit.

## Plasmid shuffling

To assess the genetic interactions between different *CFI1/NET1* phospho-mutants and FEAR mutants (*slk19Δ* or *spo12Δ*), we constructed strains in the background of *slk19Δ* or *spo12Δ* with *cfi1/net1Δ* carrying a *URA3*-based *CEN* plasmid expressing wild-type *CFI1/NET1* (pA2858) and different *CFI1/NET1* phospho-mutants integrated at the *leu2* locus (A41645-A41654). Growing these strains on plates with 5'-fluoroorotic acid (5-FOA) selects cells that are viable after losing the *URA3* plasmid.

## Data and code availability

The mass spectrometry phosphoproteomics data have been deposited to the ProteomeXchange Consortium via the PRIDE (*Perez-Riverol et al., 2019*) partner repository with the dataset identifier

PXD020369. Custom MATLAB scripts are available at https://github.com/snow-zhou/Dbf2-Mob1 (copy archived at swh:1:rev:edb372c2e4ddf8eb2278536a7fa580abaa60acf1).

## Acknowledgements

We thank F Luca (Philadelphia, USA), R Deshaies (Caltech, USA), E Schiebel (ZMBH, Germany), E Unal (Berkeley, USA), and J Haber (Waltham, USA) for strains and reagents; R Schiavoni and the Bio-polymers and Proteomics Facility at Koch Institute Swanson Biotechnology Center for processing samples for the TurboID-MS experiments; S Bell, A Seshan, and members of our laboratory for their critical reading of the manuscript. This work was supported by National Institute of General Medical Science (GM 118066 to AA) and the Eunice Kennedy Shriver National Institute of Child Health and Human Development (HD085866 to AA). XZ was supported by a Helen Hay Whitney postdoctoral fellowship. YL was supported in part by a Pilot Grant from Yale Cancer Center. AA was also investigator of the Howard Hughes Medical Institute and the Paul F Glenn Center for Biology of Aging Research at MIT.

## Additional information

### Funding

| Funder | Grant reference number | Author |
|---|---|---|
| National Institute of General Medical Sciences | GM118066 | Angelika Amon |
| Eunice Kennedy Shriver National Institute of Child Health and Human Development | HD085866 | Angelika Amon |
| Howard Hughes Medical Institute | | Angelika Amon |
| Paul F. Glenn Center for Biology of Aging Research at MIT | | Angelika Amon |
| Helen Hay Whitney Foundation | | Xiaoxue Zhou |
| Yale Cancer Center | Pilot Award | Yansheng Liu |

The funders had no role in study design, data collection and interpretation, or the decision to submit the work for publication.

### Author contributions

Xiaoxue Zhou, Conceptualization, Formal analysis, Investigation, Methodology, Writing - original draft; Wenxue Li, Yansheng Liu, Investigation, Methodology, Writing - review and editing; Angelika Amon, Conceptualization, Supervision, Funding acquisition, Writing - original draft, Project administration

### Author ORCIDs

Xiaoxue Zhou https://orcid.org/0000-0002-4551-0608
Yansheng Liu http://orcid.org/0000-0002-2626-3912
Angelika Amon https://orcid.org/0000-0001-9837-0314

### Decision letter and Author response

Decision letter https://doi.org/10.7554/eLife.63645.sa1
Author response https://doi.org/10.7554/eLife.63645.sa2

## Additional files

### Supplementary files

• Supplementary file 1. Yeast strains used in this study.

- Supplementary file 2. Plasmids used in this study.
- Supplementary file 3. Summary of TurboID labeling experiments.
- Supplementary file 4. Phosphorylation of Cfi1/Net1.
- Supplementary file 5. Summary of phosphoproteomics results for inhibiting Cdc15 and Cdc5 in anaphase cells and Cdc5 in metaphase cells.
- Supplementary file 6. Complete list of phosphopeptides for each phosphorylation site detected in the DIA-MS experiments.
- Transparent reporting form

## Data availability

The mass spectrometry phosphoproteomics data have been deposited to the ProteomeXchange Consortium via the PRIDE partner repository with the dataset identifier PXD020369. Custom MAT-LAB scripts are available at https://github.com/snow-zhou/Dbf2-Mob1. (Copy archived at https://archive.softwareheritage.org/swh:1:rev:edb372c2e4ddf8eb2278536a7fa580abaa60acf1/) All data generated or analyzed during this study are included in the manuscript and supporting files.

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
