## [Decision Letter]

**Acceptance summary:**

Signal propagation across compartments underlies many cellular decisions. Using the paradigm of mitotic exit in budding yeast, the authors elegantly combine optogenetics, proximity labelling and phosphoproteomics to understand cross-compartment signal generation. They reveal how a signal at the yeast spindle pole body is communicated to the nucleolus resulting in activation of a phosphatase which elicits mitotic exit.

**Decision letter after peer review:**

Thank you for submitting your article "Cross-compartment signal propagation in the Mitotic Exit Network" for consideration by *eLife*. Your article has been reviewed by three peer reviewers, including Adèle L Marston as the Reviewing Editor and Reviewer #1, and the evaluation has been overseen by Anna Akhmanova as the Senior Editor.

The reviewers have discussed the reviews with one another and the Reviewing Editor has drafted this decision to help you prepare a revised submission.

We would like to draw your attention to changes in our revision policy that we have made in response to COVID-19 (https://elifesciences.org/articles/57162). Specifically, we are asking editors to accept without delay manuscripts, like yours, that they judge can stand as *eLife* papers without additional data, even if they feel that they would make the manuscript stronger. Therefore the revisions requested below only address clarity and presentation. Although suggestions for experiments that will address specific points are made, appropriate textual /presentation changes and/or re-analysis of existing data will be sufficient.

Summary:

Zhou et al. show how a signal generated at the yeast spindle pole body is relayed to the nucleolus to elicit mitotic exit. Using elegantly designed experiments and innovative tools (proximity labelling, light-induced dimerization, phosphoproteomics) they cleanly demonstrate that following its activation on the SPB, Mob1-Dbf2 kinase complex translocates into the nucleolus during anaphase. Polo kinase (Cdc5) phosphorylates Cfi1/Net1 in the nucleolus already in metaphase priming it for docking of Mob1-Dbf2 upon its nuclear translocation in anaphase. Mob1-Dbf2, in turn, phosphorylates Cfi1/Net1, freeing Cdc14 phosphatase to allow mitotic exit. This manuscript contains an impressive amount of data and complementary approaches which tie together into a very convincing and compelling narrative. The result is an answer to the long-standing question of how mitotic exit is molecularly initiated by the mitotic exit network, with important implications for understanding the spatial organisation of signalling networks more broadly. There are also a number of findings and resources that are of general interest that go beyond the main message of this study, such as the Cdc5- and Cdc15-dependent phosphoproteome datasets and the finding that Dbf2-Mob1 is normally kept out of the nucleus by Crm through the NES of Dbf2. The manuscript is suitable for publication after addressing the following points:

Essential revisions:

1) The effects of Mob1 overexpression on mitotic exit and the consequences of this on the ability to accurately quantify its nuclear enrichment should be clarified. The N-terminally truncated Mob1Δ78 and Mob1Δ132 proteins are expressed at significantly higher levels than wild type Mob1 (Figure 1—figure supplement 1A). Suppression of the *cdc15-2* and *tem1-3* MEN mutants by these Mob1 truncations is argued not to be a consequence of their increased expression levels, since Mob1 overexpression from the GPD promoter did not suppress the growth defect of *cdc15-2* or *tem1-2* cells. Does overexpression of Mob1Δ78 and Mob1Δ132 under the GPD promoter still rescue the MEN mutants? A comparison of timing of NLS_Cdc14_ release in cells with GPD-Mob1 and endogenous Mob1 would show any effect of overexpression on mitotic exit.

In addition, Mob1 overexpression is indicated not to increase nucleolar localization of the protein. However, this is difficult to evaluate in Figure 1—figure supplement 1C due to the elevated nuclear background. A methodology like the BiFC assay (Sung et al., 2007) could also be helpful to get rid of this background signal. Alternatively, nucleolar enrichment quantifications could be done by normalizing the Mob1 nucleolar signal against Mob1 nuclear signal for this experiment. It is noted, however, that this concern does not change the conclusions about nucleolar enrichment of endogenous or truncated forms of Mob1 as there nuclear background is not observed with the Mob1 truncated mutants, despite their increased expression. Additionally, the Δ132 which is less expressed than the Δ78 has a more significant phenotype in nucleolar enrichment.

2) The Mob1Δ132 truncated protein is significantly enriched in the nucleolus in anaphase (Figure 1F). Given that Dbf2-Mob1 needs to be mobilized from the SPBs into the nucleus to release Cdc14 and that the authors demonstrate that Mob1Δ132 is a hyper-activated form of Mob1, it is surprising that dissociation of this truncated Mob1 protein from the SPBs is nonetheless delayed (Figure 1E). However, this delay almost disappears in Figure 4A. The authors should clarify the timing of truncated Mob1 dissociation from the SPB with respect to mitotic exit. For example, the provision of spindle elongation graphs would allow any delay in mitotic exit or spindle elongation phenotypes to be observed for each experiment. The definition of anaphase onset (spindle length >3 um) and the alignment of their data accordingly inherently may affect the appearance of these plots especially in the presence of prior mitotic arrest or spindle phenotypes.

Likewise, it is also surprising that the Dbf2-L12A protein, despite being accumulated in the nucleus and displaying an increase of at least 40% in its nucleolar localization, showed normal dynamics of association and enrichment levels on the SPBs (Figure 3E). Shouldn't the levels of Dbf2-Mob1 complex on the SPBs also be expected to decrease in this case?

3) The localization of eGFP-tagged Mob1 to the nucleolus is hard to assess when expressed at endogenous levels. The authors claim that Mob1-eGFP is transiently mobilized to the nucleolus only after segregation of the rDNA in late anaphase (Figure 1C). However, in that particular experiment the protein in fact seems to localize to the nucleolus before it is indicated. In contrast, nucleolar localization of Mob1 appears to occur after Cdc14 release in Figure 1—figure supplement 2E. Could the authors try a different approach to more precisely characterize the intracellular localization of Mob1 at endogenous levels? In this sense, a possibility might be to use the BiFC assay. By tagging Mob1 and Cfi1 with the N- and C-terminal parts of the truncated yellow-fluorescent Venus protein, the reconstitution of fluorescence through the interaction of Mob1 and Cfi1 could more neatly indicate in vivo when the Dbf2-Mob1 complex is translocated into the nucleolus. Alternatively, the authors could present a quantification of both *mob1* nucleolar and *cdc14* nucleolar signals for this data.

4) The manuscript would benefit from further introduction and discussion of the phosphorylation site data in relation to what was already known for readers that are less familiar with the MEN. Specific points related to this are:

a) Could the authors briefly summarize the previous data on Cdc5 phosphorylation of Cfi1/Net1 and Dbf2/Mob1 phosphorylation of Cfi1/Net1 (i.e. Loughrey Chen et al., 2002, Shou et al., 2002, Yoshida and Toh-e, 2002, Mah et al., 2005) in the Introduction section?

b) The previously identified phosphorylation sites S17 and S20 within the NES of Dbf2 found to be phosphorylated in cells arrested in late anaphase through overexpression of Clb2∆DB (Holt et al., 2009) appear not to have been identified in the current study. Is this a technical problem (sites were in a region not covered in the proteomic analysis) or biological (phosphorylation of these sites is not Cdc15-dependent)? The authors also observe that the *dbf2-s17,20A* mutant exacerbated the temperature sensitivity of *cdc15-2*. However the same is also true for *dbf2-s17,20D* (Figure 3—figure supplement 3A). Given that *dbf2-s17,20D* disrupted the NES (Figure 3—figure supplement 2E) how do authors explain *dbf2-s17,20D* exacerbating the temperature sensitivity of *cdc15-2*? Do the authors have any other evidence to show that these phosphorylations are *cdc15*-dependent?

---

## [Author Response]

Essential revisions:1) The effects of Mob1 overexpression on mitotic exit and the consequences of this on the ability to accurately quantify its nuclear enrichment should be clarified. The N-terminally truncated Mob1Δ78 and Mob1Δ132 proteins are expressed at significantly higher levels than wild type Mob1 (Figure 1—figure supplement 1A). Suppression of the cdc15-2 and tem1-3 MEN mutants by these Mob1 truncations is argued not to be a consequence of their increased expression levels, since Mob1 overexpression from the GPD promoter did not suppress the growth defect of cdc15-2 or tem1-2 cells. Does overexpression of Mob1Δ78 and Mob1Δ132 under the GPD promoter still rescue the MEN mutants? A comparison of timing of NLS_Cdc14_ release in cells with GPD-Mob1 and endogenous Mob1 would show any effect of overexpression on mitotic exit.

To clarify the effect of Mob1 overexpression on mitotic exit, we have compared the kinetics of NLS_Cdc14_ release and anaphase progression in cells with pGPD-Mob1 and endogenous Mob1 (Figure 1—figure supplement 1E) and showed that overexpression of Mob1 from the *GPD* promoter does not accelerate mitotic exit. In fact, pGPD-Mob1 slightly delayed MEN activation (release of NLS_Cdc14_ reporter) and mitotic exit (spindle breakdown). This is consistent with our observation that pGPD-Mob1 displayed small negative genetic interactions with *cdc15-2* and *tem1-3* (Figure 1—figure supplement 1A). Furthermore, a previous study (Rock et al., 2013) which overexpressed Mob1 from the inducible *GAL1-10* protomer also found that Mob1 overexpression does not accelerate mitotic exit or Cdc14 release. We conclude that overexpression of Mob1 is not hyperactive and the effects we observed with the N-terminally truncated forms of Mob1 were not due to overexpression. We have not tried to overexpress Mob1Δ78 and Mob1Δ132 under the *GPD* promoter. Given the slight negative effect of full-length Mob1 under overexpression, we reasoned that overexpression and truncation would likely have opposing effects on Mob1 activity making the results difficult to interpret.

In addition, Mob1 overexpression is indicated not to increase nucleolar localization of the protein. However, this is difficult to evaluate in Figure 1—figure supplement 1C due to the elevated nuclear background. A methodology like the BiFC assay (Sung et al., 2007) could also be helpful to get rid of this background signal. Alternatively, nucleolar enrichment quantifications could be done by normalizing the Mob1 nucleolar signal against Mob1 nuclear signal for this experiment. It is noted, however, that this concern does not change the conclusions about nucleolar enrichment of endogenous or truncated forms of Mob1 as there nuclear background is not observed with the Mob1 truncated mutants, despite their increased expression. Additionally, the Δ132 which is less expressed than the Δ78 has a more significant phenotype in nucleolar enrichment.

To address the issue with elevated nuclear background for pGPD-Mob1, we have included quantification of the nucleolar enrichment relative to the nuclear enrichment of pGPD-Mob1 in comparison with Mob1Δ132 (Figure 1—figure supplement 1D). In contrast to the clear enrichment of Mob1Δ132 in the nucleolus over nucleus, pGPD-Mob1 displayed similar intensities in the nucleolus and nucleus during anaphase. These data support our assessment that the increased nucleolar localization of Mob1Δ78 and Mob1Δ132 is not due to their elevated expression levels. Rather, we propose that the increase in nucleolar localization results from the elimination of the autoinhibitory N-terminus of Mob1.

2) The Mob1Δ132 truncated protein is significantly enriched in the nucleolus in anaphase (Figure 1F). Given that Dbf2-Mob1 needs to be mobilized from the SPBs into the nucleus to release Cdc14 and that the authors demonstrate that Mob1Δ132 is a hyper-activated form of Mob1, it is surprising that dissociation of this truncated Mob1 protein from the SPBs is nonetheless delayed (Figure 1E). However, this delay almost disappears in Figure 4A. The authors should clarify the timing of truncated Mob1 dissociation from the SPB with respect to mitotic exit. For example, the provision of spindle elongation graphs would allow any delay in mitotic exit or spindle elongation phenotypes to be observed for each experiment. The definition of anaphase onset (spindle length >3 um) and the alignment of their data accordingly inherently may affect the appearance of these plots especially in the presence of prior mitotic arrest or spindle phenotypes.

Localization of Mob1 and truncated forms of Mob1 to SPBs and the nucleolus are correlated temporally with each other and with MEN activation (Figure 1—figure supplement 2C-D). The “dissociation from SPBs” we referred to in the manuscript was meant to describe the decline of Dbf2-Mob1’s localization (or net dissociation) at SPBs. This net dissociation occurs towards the end of anaphase when cells exit from mitosis and reflects inactivation of the pathway (likely a result of dephosphorylation of Cdc15-phosphorylated Nud1, Dbf2-Mob1’s docking site at SPB). This net dissociation should not be confused with the dynamic association and dissociation of individual Dbf2-Mob1 molecules at SPBs that presumably precedes their localization to the nucleolus. The dynamic association and dissociation of Dbf2-Mob1 at SPBs occurs throughout the time when Dbf2-Mob1 is observed to localize at SPBs as shown by the FRAP analysis (Figure 1B). We now realized that our usage of the word “dissociation” in the original text was confusing. Given that the statement we made regarding Mob1Δ132 is not an important argument for the paper, we have removed this sentence. We have also included graphs for spindle elongation for all Mob1 alleles in Figure 1—figure supplement 1F.

Likewise, it is also surprising that the Dbf2-L12A protein, despite being accumulated in the nucleus and displaying an increase of at least 40% in its nucleolar localization, showed normal dynamics of association and enrichment levels on the SPBs (Figure 3E). Shouldn't the levels of Dbf2-Mob1 complex on the SPBs also be expected to decrease in this case?

The amounts of Dbf2-Mob1 at SPBs and in the nucleolus at any given time are relatively small compared to the cytoplasmic pool. No noticeable depletion of Dbf2-Mob1 in the cytosol was observed in anaphase when Dbf2-Mob1 is enriched at the SPBs and in the nucleolus (see Author response image 1). Furthermore, we did not observe significant changes in cytoplasmic concentration of Mob1 when expressing the *dbf2-L12A* mutant despite the notable increase in nucleolar localization (Author response image 1). The lack of change in Mob1 intensity in the cytosol upon anaphase onset and with *dbf2-L12A* was not due to an inability to detect changes in Mob1 cytoplasmic concentration. When we recruited Dbf2-Mob1 to the SPBs or mitochondria surface (via ectopically expressed Spc72 or Tom7 anchored PhyB) using optogenetics (Figure 3—figure supplement 1B), we observed clear depletion of Mob1 from the cytosol (quantified in Author response image 1), particularly for PhyB-Tom7 likely due to the large surface area of mitochondria relative to SPBs. Rather, we think these observations revealed that the cytoplasmic pool of Mob1 is large relative to the amount of Mob1 that can bind to SPBs and are transported into the nucleus. Given that the SPB pool of Dbf2-Mob1 is in a dynamic equilibrium with the cytosol, we reason that moderate changes in the nuclear/nucleolar Mob1 level will not have any major effect on Mob1 localization at SPBs.

One potential exception to this explanation is the reduction in dSPB localization of Mob1 observed when Cfi1 was overexpressed from the strong *GAL1/10* promoter (Figure 2F). However, this reduction in dSPB localization was not due to the increased localization in the nucleolus (note that the Mob1 truncation mutants that showed a higher increase in nucleolar localization due to Cfi1 overexpression showed less reduction in dSPB localization (Figure 2F)). Rather, the reduced dSPB localization was likely a result of Cdc14 inhibition by Cfi1 overexpression (Visintin et al., 1999), similar to what we observed with *cdc14-3* (Campbell et al., 2019; the original Figure 4—figure supplement 1B). The effect of Cdc14 inhibition under Cfi1 overexpression was also demonstrated by the defect in MEN activation as illustrated with the MEN activity reporter NLS_Cdc14_ (Author response image 1). We have included a brief clarification in the main text (subsection “Dbf2-Mob1 localizes to the nucleolus through interacting with Cfi1/Net1”).

**Author response image 1. sa2fig1:** Changes in the nucleolar localization of Dbf2-Mob1 do not affect its SPB enrichment. (A) Raw intensity measurements for the experiments shown in Figure 3E. (B) Quantification of Mob1 intensity in the cytosol or PhyB anchored region upon PhyB activation for the optogenetics experiments shown in Figure 3—figure supplement 1B. (C) Kinetics of MEN activation (as measured by the release of NLS_Cdc14_ reporter) for the experiments shown in Figure 2F with *GAL-CFI1*. For all graphs, solid lines represent the average of single cell traces aligned to the event described in the axis. Shaded areas represent 95% confidence intervals.

3) The localization of eGFP-tagged Mob1 to the nucleolus is hard to assess when expressed at endogenous levels. The authors claim that Mob1-eGFP is transiently mobilized to the nucleolus only after segregation of the rDNA in late anaphase (Figure 1C). However, in that particular experiment the protein in fact seems to localize to the nucleolus before it is indicated. In contrast, nucleolar localization of Mob1 appears to occur after Cdc14 release in Figure 1—figure supplement 2E. Could the authors try a different approach to more precisely characterize the intracellular localization of Mob1 at endogenous levels? In this sense, a possibility might be to use the BiFC assay. By tagging Mob1 and Cfi1 with the N- and C-terminal parts of the truncated yellow-fluorescent Venus protein, the reconstitution of fluorescence through the interaction of Mob1 and Cfi1 could more neatly indicate in vivo when the Dbf2-Mob1 complex is translocated into the nucleolus. Alternatively, the authors could present a quantification of both mob1 nucleolar and cdc14 nucleolar signals for this data.

Given the transient and subtle nature of the nucleolar localization of wild-type Mob1 observed by fluorescence microscopy, a more sensitive assay like BiFC would be helpful. However, there are several issues with using the BiFC assay for our system: (1) The BiFC complementation is irreversible creating a constitutive fusion between Mob1 and Cfi1. Such a situation would lead to high background from interactions of previous cell cycles. Alternatively, expressing an inducible fusion would suffer from the challenges of controlling the expression level. Additionally, trapping Dbf2-Mob1 in the nucleolus with the complementation might lead to defects in the MEN and cell cycle progression; (2) There is a notable delay between the time when the fusion proteins interact with each other and the time when the complex becomes fluorescent ranging from minutes to hours (Kerppola T.K., Annu Rev Biophys. 2008; 37: 465–487). This limit of temporal resolution is problematic given that the nucleolar localization we observed lasted ~10 mins within the 20-min-long anaphase (Figure 1E); (3) The success of complementation depends on the orientation of the two interacting proteins relative to the fragments and might not work in our case given that Dbf2-Mob1 interacts with the middle segment (zone 2) of the very large Cfi1 protein.

To clarify the relative timing of Cdc14 release and Dbf2-Mob1’s nucleolar localization, we have labeled the two different stages of Cdc14 release in Figure 1—figure supplement 2E. As the reviewers pointed out, nucleolar localization of Mob1 occurs after Cdc14 release in Figure 1—figure supplement 2E, We believe this Cdc14 release before nucleolar localization of Mob1 is the early anaphase release mediated by the FEAR network (labeled on the top row of Figure 1—figure supplement 2E). The FEAR network promotes Cdc14 release in early anaphase while the MEN drives Cdc14 release in late anaphase. We found that Dbf2-Mob1 localizes to the nucleolus in late anaphase (peak between 10 to 20 mins after anaphase onset, Figure 1E), typically after segregation of the nucleolus (occurs ~9 mins after anaphase onset on average, Figure 6C). In contrast, Cdc14 is first released by the FEAR network upon anaphase onset (Figure 1—figure supplement 2E and quantified in Figure 6A) before the peak of Dbf2-Mob1’s nucleolar localization. This timing of nucleolar Dbf2-Mob1 we observed is consistent with the previously established role of the MEN in releasing Cdc14 in late anaphase.

4) The manuscript would benefit from further introduction and discussion of the phosphorylation site data in relation to what was already known for readers that are less familiar with the MEN. Specific points related to this are:a) Could the authors briefly summarize the previous data on Cdc5 phosphorylation of Cfi1/Net1 and Dbf2/Mob1 phosphorylation of Cfi1/Net1 (i.e. Loughrey Chen et al., 2002, Shou et al., 2002, Yoshida and Toh-e, 2002, Mah et al., 2005) in the Introduction section?

We have added a brief summary of literature on phosphorylation of Cfi1/Net1 by Cdc5 (previously described in the Results section) and Dbf2-Mob1 in the Introduction.

b) The previously identified phosphorylation sites S17 and S20 within the NES of Dbf2 found to be phosphorylated in cells arrested in late anaphase through overexpression of Clb2∆DB (Holt et al., 2009) appear not to have been identified in the current study. Is this a technical problem (sites were in a region not covered in the proteomic analysis) or biological (phosphorylation of these sites is not Cdc15-dependent)? The authors also observe that the dbf2-s17,20A mutant exacerbated the temperature sensitivity of cdc15-2. However the same is also true for dbf2-s17,20D (Figure 3—figure supplement 3A). Given that dbf2-s17,20D disrupted the NES (Figure 3—figure supplement 2E) how do authors explain dbf2-s17,20D exacerbating the temperature sensitivity of cdc15-2? Do the authors have any other evidence to show that these phosphorylations are cdc15-dependent?

Unfortunately, we did not detect any phosphopeptides of Dbf2 containing S17 or S20 in our phosphoproteomic analysis of anaphase cultures with either *WT* cells or *cdc15-as1* cells. We think that this is likely a technical problem with MS sample preparation and detection. As a result, we could not determine whether these phosphorylations are Cdc15-dependent.

It is indeed intriguing but not unexpected that *dbf2-S17,20D* exacerbated the temperature sensitivity of *cdc15-2* above 30°C similar to *dbf2-S17,20A*. The simplest explanation for this observation is that permanently disrupting the NES with *dbf2-S17,20D* to mimic constitutive phosphorylation is not the same as having cell-cycle-regulated and potentially partial/dynamic phosphorylation of the NES. Maintaining a dynamic shuttling of Dbf2-Mob1 between the nucleus and cytosol might be important for Dbf2-Mob1’s activation and function, particularly when Cdc15 level is reduced. In addition, our results with *dbf2-S17,20A* which retained anaphase specific increase in nuclear access of Dbf2-Mob1 indicates that there are regulatory mechanism(s) other than phosphorylation of the NES that impact Dbf2-Mob1 localization. Thus, *dbf2-S17,20D* is not expected to bypass the proposed role of Cdc15 in regulating Dbf2-Mob1’s nuclear access in anaphase and might even lead to additional perturbations. We also note that at 30°C, there is a noticeable difference between *dbf2-S17,20D* and *dbf2-S17,20A* in *cdc15-2* suppression: the former grew slightly better than *WTDBF2* (larger colonies) whereas the latter did not. However, given that we do not have additional evidence to show these Dbf2 phosphorylation events are Cdc15-dependent in the current study and this is not a major focus of our story, we have moved the description of this result to the Discussion where we propose a possible mechanism for regulating Dbf2-Mob1’s nuclear access through phosphorylating Dbf2’s NES by Cdc15.